# POLICY GRADIENT WITH TREE EXPANSION

## ABSTRACT

Policy gradient methods are notorious for having a large variance and high sample complexity. To mitigate this, we introduce SoftTreeMax—a generalization of softmax that employs planning. In SoftTreeMax, we extend the traditional logits with the multi-step discounted cumulative reward, topped with the logits of future states. We analyze SoftTreeMax and explain how tree expansion helps to reduce its gradient variance. We prove that the variance depends on the chosen tree-expansion policy. Specifically, we show that the closer the induced transitions are to being state-independent, the stronger the variance decay. With approximate forward models, we prove that the resulting gradient bias diminishes with the approximation error while retaining the same variance reduction. Ours is the first result to bound the gradient bias for an approximate model. In a practical implementation of SoftTreeMax, we utilize a parallel GPU-based simulator for fast and efficient tree expansion. Using this implementation in Atari, we show that SoftTreeMax reduces the gradient variance by three orders of magnitude. This leads to better sample complexity and improved performance compared to distributed PPO.

## 1 INTRODUCTION

Policy Gradient (PG) methods (Sutton et al., 1999) for Reinforcement Learning (RL) are often the first choice for environments that allow numerous interactions at a fast pace (Schulman et al., 2017). Their success is attributed to several factors: they are easy to distribute to multiple workers, require no assumptions on the underlying value function, and have both on-policy and off-policy variants.

Despite these positive features, PG algorithms are also notoriously unstable due to the high variance of the gradients computed over entire trajectories (Liu et al., 2020; Xu et al., 2020). As a result, PG algorithms tend to be highly inefficient in terms of sample complexity. Several solutions have been proposed to mitigate the high variance issue, including baseline subtraction (Greensmith et al., 2004; Thomas & Brunskill, 2017; Wu et al., 2018), anchor-point averaging (Papini et al., 2018), and other variance reduction techniques (Zhang et al., 2021; Shen et al., 2019; Pham et al., 2020).

A second family of algorithms that achieved state-of-the-art results in several domains is based on planning. Planning is exercised primarily in the context of value-based RL and is usually implemented using a Tree Search (TS) (Silver et al., 2016; Schrittwieser et al., 2020). In this work, we combine PG with TS by introducing a parameterized differentiable policy that incorporates tree expansion. Namely, our SoftTreeMax policy replaces the standard policy logits of a state and action, with the expected value of trajectories that originate from these state and action. We consider two variants of SoftTreeMax, one for cumulative reward and one for exponentiated reward.

Combining TS and PG should be done with care given the biggest downside of PG—its high gradient variance. This raises questions that were ignored until this work: (i) How to design a PG method based on tree-expansion that is stable and performs well in practice? and (ii) How does the tree-expansion policy affect the PG variance? Here, we analyze SoftTreeMax, and provide a practical methodology to choose the expansion policy to minimize the resulting variance. Our main result shows that a desirable expansion policy is one, under which the induced transition probabilities are similar for each starting state. More generally, we show that the gradient variance of SoftTreeMax decays at a rate of $|\lambda_2|^d$, where $d$ is the depth of the tree and $\lambda_2$ is the second eigenvalue of the transition matrix induced by the tree expansion policy. This work is the first to prove such a relation between PG variance and tree expansion policy. In addition, we prove that the with an approximate forward model, the bias of the gradient is bounded proportionally to the approximation error of the model.

To verify our results, we implemented a practical version of SoftTreeMax that exhaustively searches the entire tree and applies a neural network on its leaves. We test our algorithm on a parallelized Atari GPU simulator (Dalton et al., 2020). To enable a tractable deep search, up to depth eight, we also introduce a pruning technique that limits the width of the tree. We do so by sampling only the most promising nodes at each level. We integrate our SoftTreeMax GPU implementation into the popular PPO (Schulman et al., 2017) and compare it to the flat distributed variant of PPO. This allows us to demonstrate the potential benefit of utilizing learned models while isolating the fundamental properties of TS without added noise. In all tested Atari games, our results outperform the baseline and obtain up to 5x more reward. We further show in Section 6 that the associated gradient variance is smaller by three orders of magnitude in all games, demonstrating the relation between low gradient variance and high reward.

We summarize our key contributions. (i) We show how to combine two families of SoTA approaches: PG and TS by **introducing SoftTreeMax:** a novel parametric policy that generalizes softmax to planning. Specifically, we propose two variants based on cumulative and exponentiated rewards. (ii) We **prove that the gradient variance of SoftTreeMax in its two variants decays** with its tree depth. Our analysis sheds new light on the choice of tree expansion policy. It raises the question of optimality in terms of variance versus the traditional regret; e.g., in UCT (Kocsis & Szepesvári, 2006). (iii) We prove that with an approximate forward model, the **gradient bias is proportional to the approximation error**, while retaining the variance decay. This quantifies the accuracy required from a learned forward model. (iv) We **implement a differentiable deep version of SoftTreeMax** that employs a parallelized GPU tree expansion. We demonstrate how its gradient variance is reduced by three orders of magnitude over PPO while obtaining up to 5x reward.

## 2 PRELIMINARIES

Let $\Delta_U$ denote simplex over the set $U$. Throughout, we consider a discounted Markov Decision Process (MDP) $\mathcal{M} = (\mathcal{S}, \mathcal{A}, P, r, \gamma, \nu)$, where $\mathcal{S}$ is a finite state space of size $S$, $\mathcal{A}$ is a finite action space of size $A$, $r : \mathcal{S} \times \mathcal{A} \to [0, 1]$ is the reward function, $P : \mathcal{S} \times \mathcal{A} \to \Delta_{\mathcal{S}}$ is the transition function, $\gamma \in (0, 1)$ is the discount factor, and $\nu \in \mathbb{R}^S$ is the initial state distribution. We denote the transition matrix starting from state $s$ by $P_s \in [0, 1]^{A \times S}$, i.e., $[P_s]_{a,s'} = P(s'|a, s)$. Similarly, let $R_s = r(s, \cdot) \in \mathbb{R}^A$ denote the corresponding reward vector. Separately, let $\pi : \mathcal{S} \to \Delta_{\mathcal{A}}$ be a stationary policy. Let $P^\pi$ and $R_\pi$ be the induced transition matrix and reward function, respectively, i.e., $P^\pi(s'|s) = \sum_a \pi(a|s) \Pr(s'|s, a)$ and $R_\pi(s) = \sum_a \pi(a|s) r(s, a)$. Denote the stationary distribution of $P^\pi$ by $\mu_\pi \in \mathbb{R}^S$ s.t. $\mu_\pi^\top P^\pi = P^\pi$, and the discounted state visitation frequency by $d_\pi$ so that $d_\pi^\top = (1 - \gamma) \sum_{t=0}^\infty \gamma^t \nu^\top (P^\pi)^t$. Also, let $V^\pi \in \mathbb{R}^S$ be the value function of $\pi$ defined by $V^\pi(s) = \mathbb{E}^\pi \left[ \sum_{t=0}^\infty \gamma^t r(s_t, \pi(s_t)) \mid s_0 = s \right]$, and let $Q^\pi \in \mathbb{R}^{S \times A}$ be the Q-function such that $Q^\pi(s, a) = \mathbb{E}^\pi [r(s, a) + \gamma V^\pi(s')]$. Our goal is to find an optimal policy $\pi^\star$ such that $V^\star(s) \equiv V^{\pi^\star}(s) = \max_\pi V^\pi(s), \ \forall s \in \mathcal{S}$.

For the analysis in Section 4, we introduce the following notation. Denote by $\Theta \in \mathbb{R}^S$ the vector representation of $\theta(s) \ \forall s \in \mathcal{S}$. For a vector $u$, denote by $\exp(u)$ the coordinate-wise exponent of $u$ and by $D(u)$ the diagonal square matrix with $u$ in its diagonal. For a matrix $A$, denote its $i$-th eigenvalue by $\lambda_i(A)$. Denote the $k$-dimensional identity matrix and all-ones vector by $I_k$ and $\mathbf{1}_k$, respectively. Also, denote the trace operator by $\mathrm{Tr}$. Finally, we treat all vectors as column vectors.

### 2.1 POLICY GRADIENT

PG schemes seek to maximize the cumulative reward as a function of the policy $\pi_\theta(a|s)$ by performing gradient steps on $\theta$. The celebrated Policy Gradient Theorem (Sutton et al., 1999) states that

$$\frac{\partial}{\partial \theta} \nu^\top V^{\pi_\theta} = \mathbb{E}_{s \sim d_{\pi_\theta}, a \sim \pi_\theta(\cdot|s)} \left[ \nabla_\theta \log \pi_\theta(a|s) Q^{\pi_\theta}(s, a) \right],$$

where $\nu$ and $d_{\pi_\theta}^\top$ are as defined above. The variance of the gradient is thus

$$\mathrm{Var}_{s \sim d_{\pi_\theta}, a \sim \pi_\theta(\cdot|s)} \left( \nabla_\theta \log \pi_\theta(a|s) Q^{\pi_\theta}(s, a) \right). \tag{1}$$

In the notation above, we denote the variance of a vector random variable $X$ by

$$\mathrm{Var}_x (X) = \mathrm{Tr} \left[ \mathbb{E}_x \left[ (X - \mathbb{E}_x X)^\top (X - \mathbb{E}_x X) \right] \right],$$

similarly as in (Greensmith et al., 2004). From now on, we drop the subscript from $\mathrm{Var}$ in (1) for brevity. When the action space is discrete, a commonly used parameterized policy is softmax: $\pi_\theta(a|s) \propto \exp\left(\theta(s,a)\right)$, where $\theta : \mathcal{S} \times \mathcal{A} \to \mathbb{R}$ is a state-action parameterization.

# 3 SOFTTREEMAX: EXPONENT OF TRAJECTORIES

We introduce a new family of policies called SoftTreeMax, which are a model-based generalization of the popular softmax. We propose two variants: Cumulative (C-SoftTreeMax) and Exponentiated (E-SoftTreeMax). In both variants, we replace the generic softmax logits $\theta(s,a)$ with the score of a trajectory of horizon $d$ starting from $(s,a)$, generated by applying a behavior policy $\pi_b$. In C-SoftTreeMax, we exponentiate the expectation of the logits. In E-SoftTreeMax, we first exponentiate the logits and then only compute their expectation.

**Logits**. We define the SoftTreeMax logit $\ell_{s,a}(d;\theta)$ to be the random variable depicting the score of a trajectory of horizon $d$ starting from $(s,a)$ and following the policy $\pi_b$:

$$\ell_{s,a}(d;\theta) = \gamma^{-d}\left[\sum_{t=0}^{d-1}\gamma^t r_t + \gamma^d \theta(s_d)\right]. \tag{2}$$

In the above expression, note that $s_0 = s$, $a_0 = a$, $a_t \sim \pi_b(\cdot|s_t) \ \forall t \geq 1$, and $r_t \equiv r(s_t, a_t)$. For brevity of the analysis, we let the parametric score $\theta$ in (2) be state-based, similarly to a value function. Instead, one could use a state-action input analogous to a Q-function. Thus, SoftTreeMax can be integrated into the two types of implementation of RL algorithms in standard packages. Lastly, the preceding $\gamma^{-d}$ scales the $\theta$ parametrization to correspond to its softmax counerpart.

**C-SoftTreeMax**. Given an inverse temperature parameter $\beta$, we let C-SoftTreeMax be

$$\pi_{d,\theta}^{\mathrm{C}}(a|s) \propto \exp\left[\beta\mathbb{E}^{\pi_b}\ell_{s,a}(d;\theta)\right]. \tag{3}$$

C-SoftTreeMax gives higher weight to actions that result in higher expected returns. While standard softmax relies entirely on parametrization $\theta$, C-SoftTreeMax also interpolates a Monte-Carlo portion of the reward.

**E-SoftTreeMax**. The second operator we propose is E-SoftTreeMax:

$$\pi_{d,\theta}^{\mathrm{E}}(a|s) \propto \mathbb{E}^{\pi_b}\exp\left[(\beta\ell_{s,a}(d;\theta))\right]; \tag{4}$$

here, the expectation is taken outside the exponent. This objective corresponds to the exponentiated reward objective which is often used for risk-sensitive RL (Howard & Matheson, 1972; Fei et al., 2021; Noorani & Baras, 2021). The common risk-sensitive objective is of the form $\log\mathbb{E}[\exp(\delta R)]$, where $\delta$ is the risk parameter and $R$ is the cumulative reward. Similarly to that literature, the exponent in (4) emphasizes the most promising trajectories.

**SoftTreeMax properties**. SoftTreeMax is a natural model-based generalization of softmax. For $d = 0$, both variants above coincide since (2) becomes deterministic. In that case, for a state-action parametrization, they reduce to standard softmax. When $\beta \to 0$, both variants again coincide and sample actions uniformly (exploration). When $\beta \to \infty$, the policies become deterministic and greedily optimize for the best trajectory (exploitation). For C-SoftTreeMax, the best trajectory is defined in expectation, while for E-SoftTreeMax it is defined in terms of the best sample path.

**SoftTreeMax convergence.** Under regularity conditions, for any parametric policy, PG converges to local optima (Bhatnagar et al., 2009), and thus also SoftTreeMax. For softmax PG, asymptotic (Agarwal et al., 2021) and rate results (Mei et al., 2020b) were recently obtained, by showing that the gradient is strictly positive everywhere (Mei et al., 2020b, Lemmas 8-9). We conjecture that SoftTreeMax satisfies the same property, being a generalization of softmax, but formally proving it is subject to future work.

**SoftTreeMax gradient.** The two variants of SoftTreeMax involve an expectation taken over $S^d$ many trajectories from the root state $s$ and weighted according to their probability. Thus, during the PG training process, the gradient $\nabla_\theta \log\pi_\theta$ is calculated using a weighted sum of gradients over all reachable states starting from $s$. Our method exploits the exponential number of trajectories to reduce the variance while improving performance. Indeed, in the next section we prove that the gradient variance of SoftTreeMax decays exponentially fast as a function of the behavior policy $\pi_b$

and trajectory length $d$. In the experiments in Section 6, we also show how the practical version of SoftTreeMax achieves a significant reduction in the noise of the PG process and leads to faster convergence and higher reward.

# 4 THEORETICAL ANALYSIS

In this section, we first bound the variance of PG when using the SoftTreeMax policy. Later, we discuss how the gradient bias resulting due to approximate forward models diminishes as a function of the approximation error, while retaining the same variance decay.

We show that the variance decreases with the tree depth, and the rate is determined by the second eigenvalue of the transition kernel induced by $\pi_b$. Specifically, we bound the same expression for variance as appears in (Greensmith et al., 2004, Sec. 3.5) and (Wu et al., 2018, Sec. A, Eq. (21)). Other types of analysis could instead have focused on the estimation aspect in the context of sampling (Zhang et al., 2021; Shen et al., 2019; Pham et al., 2020). Indeed, in our implementation in Section 5, we manage to avoid sampling and directly compute the expectations in Eqs. (3) and (4). As we show later, we do so by leveraging efficient parallel simulation on the GPU in feasible run-time. In our application, due to the nature of the finite action space and quasi-deterministic Atari dynamics (Bellemare et al., 2013), our expectation estimator is noiseless. We encourage future work to account for the finite-sample variance component. We defer all the proofs to Appendix A.

We begin with a general variance bound that holds for *any* parametric policy.

**Lemma 4.1** (Bound on the policy gradient variance). *Let $\nabla_\theta \log \pi_\theta(\cdot|s) \in \mathbb{R}^{A \times \dim(\theta)}$ be a matrix whose $a$-th row is $\nabla_\theta \log \pi_\theta(a|s)^\top$. For any parametric policy $\pi_\theta$ and function $Q^{\pi_\theta} : \mathcal{S} \times \mathcal{A} \to \mathbb{R}$,*

$$\mathrm{Var}\left(\nabla_\theta \log \pi_\theta(a|s) Q^{\pi_\theta}(s,a)\right) \leq \max_{s,a}\left[Q^{\pi_\theta}(s,a)\right]^2 \max_s \|\nabla_\theta \log \pi_\theta(\cdot|s)\|_F^2.$$

Hence, to bound (1), it is sufficient to bound the Frobenius norm $\|\nabla_\theta \log \pi_\theta(\cdot|s)\|_F$ for any $s$.

Note that SoftTreeMax does not reduce the gradient uniformly, which would have been equivalent to a trivial change in the learning rate. While the gradient norm shrinks, the gradient itself scales differently along the different coordinates. This scaling occurs along different eigenvectors, as a function of problem parameters ($P$, $\theta$) and our choice of behavior policy ($\pi_b$), as can be seen in the proof of the upcoming Theorem 4.4. This allows SoftTreeMax to learn a good "shrinkage" that, while reducing the overall gradient, still updates the policy quickly enough. This reduction in norm and variance resembles the idea of gradient clipping Zhang et al. (2019), where the gradient is scaled to reduce its variance, thus increasing stability and improving overall performance.

A common assumption in the RL literature (Szepesvári, 2010) that we adopt for the remainder of the section is that the transition matrix $P^{\pi_b}$, induced by the behavior policy $\pi_b$, is irreducible and aperiodic. Consequently, its second highest eigenvalue satisfies $|\lambda_2(P^{\pi_b})| < 1$.

From now on, we divide the variance results for the two variants of SoftTreeMax into two subsections. For C-SoftTreeMax, the analysis is simpler and we provide an exact bound. The case of E-SoftTreeMax is more involved and we provide for it a more general result. In both cases, we show that the variance decays exponentially with the planning horizon.

## 4.1 VARIANCE OF C-SOFTTREEMAX

We express C-SoftTreeMax in vector form as follows.

**Lemma 4.2** (Vector form of C-SoftTreeMax). *For $d \geq 1$, (3) is given by*

$$\pi_{d,\theta}^C(\cdot|s) = \frac{\exp\left[\beta\left(C_{s,d} + P_s\left(P^{\pi_b}\right)^{d-1}\Theta\right)\right]}{\boldsymbol{I}_A^\top \exp\left[\beta\left(C_{s,d} + P_s\left(P^{\pi_b}\right)^{d-1}\Theta\right)\right]}, \tag{5}$$

*where*

$$C_{s,d} = \gamma^{-d} R_s + P_s\left[\sum_{h=1}^{d-1}\gamma^{h-d}\left(P^{\pi_b}\right)^{h-1}\right]R_{\pi_b}.$$

The vector $C_{s,d} \in \mathbb{R}^A$ represents the cumulative discounted reward in expectation along the trajectory of horizon $d$. This trajectory starts at state $s$, involves an initial reward dictated by $R_s$ and an initial transition as per $P_s$. Thereafter, it involves rewards and transitions specified by $R_{\pi_b}$ and $P^{\pi_b}$, respectively. Once the trajectory reaches depth $d$, the score function $\theta(s_d)$ is applied,.

**Lemma 4.3** (Gradient of C-SoftTreeMax). *The C-SoftTreeMax gradient is given by*

$$\nabla_\theta \log \pi_{d,\theta}^C = \beta \left[ I_A - \mathbf{1}_A (\pi_{d,\theta}^C)^\top \right] P_s (P^{\pi_b})^{d-1},$$

*in $\mathbb{R}^{A \times S}$, where for brevity, we drop the $s$ index in the policy above, i.e., $\pi_{d,\theta}^C \equiv \pi_{d,\theta}^C(\cdot|s)$.*

We are now ready to present our first main result:

**Theorem 4.4** (Variance decay of C-SoftTreeMax). *For every $Q : \mathcal{S} \times \mathcal{A} \to \mathbb{R}$, the C-SoftTreeMax policy gradient variance is bounded by*

$$\mathrm{Var}\left(\nabla_\theta \log \pi_{d,\theta}^C(a|s) Q(s,a)\right) \leq 2 \frac{A^2 S^2 \beta^2}{(1-\gamma)^2} |\lambda_2(P^{\pi_b})|^{2(d-1)}.$$

We provide the full proof in Appendix A.4, and briefly outline its essence here.

*Proof outline.* Lemma 4.1 allows us to bound the variance using a direct bound on the gradient norm. The gradient is given in Lemma 4.3 as a product of three matrices, which we now study from right to left. The matrix $P^{\pi_b}$ is a row-stochastic matrix. Because the associated Markov chain is irreducible and aperiodic, it has a unique stationary distribution. This implies that $P^{\pi_b}$ has one and only one eigenvalue equal to 1; all others have magnitude strictly less than 1. Let us suppose that all these other eigenvalues have multiplicity 1 (the general case with repeated eigenvalues can be handled via Jordan decompositions as in (Pelletier, 1998, Lemma1)). Then, $P^{\pi_b}$ has the spectral decomposition $P^{\pi_b} = \mathbf{1}_S \mu_{\pi_b}^\top + \sum_{i=2}^{S} \lambda_i v_i u_i^\top$, where $\lambda_i$ is the $i$-th eigenvalue of $P^{\pi_b}$ (ordered in descending order according to their magnitude) and $u_i$ and $v_i$ are the corresponding left and right eigenvectors, respectively, and therefore $(P^{\pi_b})^{d-1} = \mathbf{1}_S \mu_{\pi_b}^\top + \sum_{i=2}^{S} \lambda_i^{d-1} v_i u_i^\top$.

The second matrix in the gradient relation in Lemma 4.3, $P_s$, is a rectangular transition matrix that translates the vector of all ones from dimension $S$ to $A : P_s \mathbf{1}_S = \mathbf{1}_A$. Lastly, the first matrix $\left[ I_A - \mathbf{1}_A (\pi_{d,\theta}^C)^\top \right]$ is a projection whose null-space includes the vector $\mathbf{1}_A$, i.e., $\left[ I_A - \mathbf{1}_A (\pi_{d,\theta}^C)^\top \right] \mathbf{1}_A = 0$. Combining the three properties above when multiplying the three matrices of the gradient, it is easy to see that the first term in the expression for $(P^{\pi_b})^{d-1}$ gets canceled, and we are left with bounded summands scaled by $\lambda_i (P^{\pi_b})^{d-1}$. Recalling that $|\lambda_i(P^{\pi_b})| < 1$ and that $|\lambda_2| \geq |\lambda_3| \geq \ldots$ for $i = 2, \ldots, S$, we obtain the desired result. $\qquad\square$

Theorem 4.4 guarantees that the variance of the gradient decays with $d$. More importantly, it also provides a novel insight for choosing the behavior policy $\pi_b$ as the policy that minimizes the absolute second eigenvalue of the $P^{\pi_b}$. Indeed, the second eigenvalue of a Markov chain relates to its connectivity and its rate of convergence to the stationary distribution (Levin & Peres, 2017).

**Optimal variance decay**. For the strongest reduction in variance, the behavior policy $\pi_b$ should be chosen to achieve an induced Markov chain whose transitions are state-independent. In that case, $P^{\pi_b}$ is a rank one matrix of the form $\mathbf{1}_S \mu_{\pi_b}^\top$, and $\lambda_2(P^{\pi_b}) = 0$. Then, $\mathrm{Var}\left(\nabla_\theta \log \pi_\theta(a|s) Q(s,a)\right) = 0$. Naturally, this can only be done for pathological MDPs; see Appendix C.1 for a more detailed discussion. Nevertheless, as we show in Section 5, we choose our tree expansion policy to reduce the variance as best as possible.

**Worst-case variance decay**. In contrast, and somewhat surprisingly, when $\pi_b$ is chosen so that the dynamics is deterministic, there is no guarantee that it will decay exponentially fast. For example, if $P^{\pi_b}$ is a permutation matrix, then $\lambda_2(P^{\pi_b}) = 1$, and advancing the tree amounts to only updating the gradient of one state for every action, as in the basic softmax.

## 4.2 VARIANCE OF E-SOFTTREEMAX

The proof of the variance bound for E-SoftTreeMax is similar to that of C-SoftTreeMax, but more involved. It also requires the assumption that the reward depends only on the state, i.e. $r(s,a) \equiv r(s)$. This is indeed the case in most standard RL environments such as Atari and Mujoco.

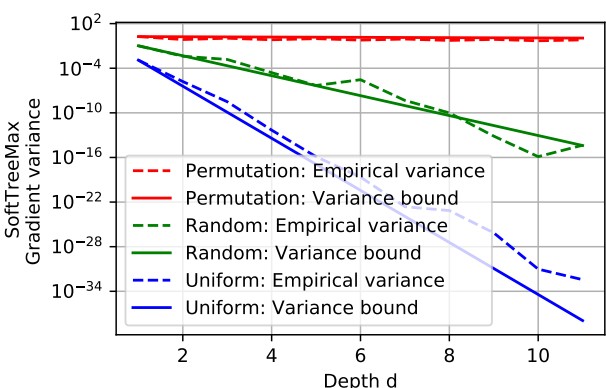

Figure 1: A comparison of the empirical PG variance and our bound for E-SoftTreeMax on randomly drawn MDPs. We present three cases for $P^{\pi_b}$: (i) close to uniform, (ii) drawn randomly, and (iii) close to a permutation matrix. This experiment verifies the optimal and worse-case rate decay cases. The variance bounds here are taken from Theorem 4.7 where we substitute $\alpha = |\lambda_2(P^{\pi_b})|$. To account for the constants, we match the values for the first point in $d = 1$.

**Lemma 4.5** (Vector form of E-SoftTreeMax). *For $d \geq 1$, (4) is given by*

$$\pi^E_{d,\theta}(\cdot|s) = \frac{E_{s,d}\exp(\beta\Theta)}{1_A^\top E_{s,d}\exp(\beta\Theta)}, \tag{6}$$

*where*

$$E_{s,d} = P_s \prod_{h=1}^{d-1} \left( D\left(\exp(\beta\gamma^{h-d}R)\right) P^{\pi_b} \right).$$

*The vector $R$ above is the $S$-dimensional vector whose $s$-th coordinate is $r(s)$.*

The matrix $E_{s,d} \in \mathbb{R}^{A \times S}$ has a similar role to $C_{s,d}$ from (5), but it represents the exponentiated cumulative discounted reward. Accordingly, it is a product of $d$ matrices as opposed to a sum. It captures the expected reward sequence starting from $s$ and then iteratively following $P^{\pi_b}$. After $d$ steps, we apply the score function on the last state as in (6).

**Lemma 4.6** (Gradient of E-SoftTreeMax). *The E-SoftTreeMax gradient is given by*

$$\nabla_\theta \log \pi^E_{d,\theta} = \beta\left[I_A - \boldsymbol{1}_A(\pi^E_{d,\theta})^\top\right] \times \frac{D\left(\pi^E_{d,\theta}\right)^{-1} E_{s,d} D(\exp(\beta\Theta))}{\boldsymbol{1}_A^\top E_{s,d}\exp(\beta\Theta)} \quad \in \quad \mathbb{R}^{A \times S},$$

*where for brevity, we drop the $s$ index in the policy above, i.e., $\pi^E_{d,\theta} \equiv \pi^E_{d,\theta}(\cdot|s)$.*

This gradient structure is harder to handle than that of C-SoftTreeMax in Lemma 4.3, but here we also can bound the decay of the variance nonetheless.

**Theorem 4.7** (Variance decay of E-SoftTreeMax). *There exists $\alpha \in (0,1)$ such that,*

$$\mathrm{Var}\left(\nabla_\theta \log \pi^E_{d,\theta}(a|s)Q(s,a)\right) \in \mathcal{O}\left(\beta^2 \alpha^{2d}\right),$$

*for every $Q$. Further, if $P^{\pi_b}$ is reversible or if the reward is constant, then $\alpha = |\lambda_2(P^{\pi_b})|$.*

**Theory versus Practice.** We demonstrate the above result in simulation. We draw a random finite MDP, parameter vector $\Theta \in \mathbb{R}^S_+$, and behavior policy $\pi_b$. We then empirically compute the PG variance of E-SoftTreeMax as given in (1) and compare it to $|\lambda_2(P^{\pi_b})|^d$. We repeat this experiment three times for different $P^{\pi_b}$: (i) close to uniform, (ii) drawn randomly, and (iii) close to a permutation matrix. As seen in Figure 1, the empirical variance and our bound match almost identically. This also suggests that $\alpha = |\lambda_2(P^{\pi_b})|$ in the general case and not only when $P^{\pi_b}$ is reversible or when the reward is constant.

### 4.3 BIAS WITH AN APPROXIMATE FORWARD MODEL

The definition of the two SoftTreeMax variants involves the knowledge of the underlying environment, in particular the value of $P$ and $r$. However, in practice, we often can only learn approximations of the dynamics from interactions, e.g., using NNs (Ha & Schmidhuber, 2018; Schrittwieser et al.,

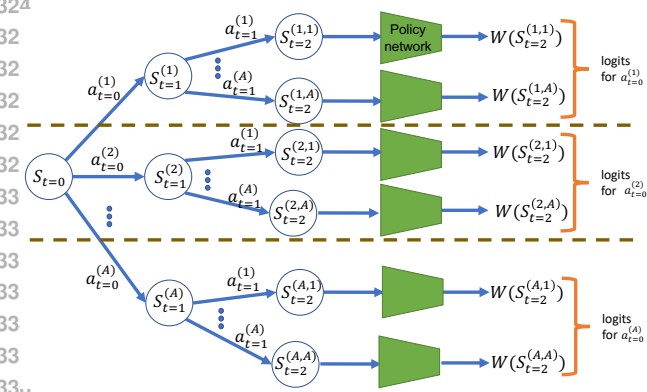

Figure 2: **SoftTreeMax policy**. Our exhaustive parallel tree expansion iterates on all actions at each state up to depth $d$ (= 2 here). The leaf state of every trajectory is used as input to the policy network. The output is then added to the trajectory's cumulative reward as described in (2). I.e., instead of the standard softmax logits, we add the cumulative discounted reward to the policy network output. This policy is differentiable and can be easily integrated into any PG algorithm. In this work, we build on PPO and use its loss function to train the policy network.

2020). Let $\hat{P}$ and $\hat{r}$ denote the approximate kernel and reward functions, respectively. In this section, we study the consequences of the approximation error on the C-SoftTreeMax gradient.

Let $\hat{\pi}_{d,\theta}^{C}$ be the C-SoftTreeMax policy defined given the approximate forward model introduced above. That is, let $\hat{\pi}_{d,\theta}^{C}$ be defined exactly as in (5), but using $\hat{R}_s, \hat{P}_s, \hat{R}_{\pi_b}$ and $\hat{P}^{\pi_b}$, instead of their unperturbed counterparts from Section 2. Then, the variance of the corresponding gradient again decays exponentially with a decay rate of $\lambda_2(\hat{P}^{\pi_b})$. However, a gradient bias is introduced. In the following, we bound this bias in terms of the approximation error and other problem parameters. The proof is provided in Appendix A.9.

**Theorem 4.8.** *Let $\epsilon$ be the maximal model mis-specification, i.e., let $\max\{\|P - \hat{P}\|, \|r - \hat{r}\|\} = \epsilon$. Then the policy gradient bias due to $\hat{\pi}_{d,\theta}^{C}$ satisfies*

$$\left\| \frac{\partial}{\partial \theta} \left( \nu^\top V^{\pi_{d,\theta}^{C}} \right) - \frac{\partial}{\partial \theta} \left( \nu^\top V^{\hat{\pi}_{d,\theta}^{C}} \right) \right\| = \mathcal{O}\left( \frac{1}{(1-\gamma)^2} S\beta^2 d\epsilon \right). \tag{7}$$

To the best of our knowledge, Theorem 4.8 is the first result that bounds the bias of the gradient of a parametric policy due to an approximate model. It states that if the learned model is accurate enough, we expect similar convergence properties for C-SoftTreeMax as we would have obtained with the true dynamics. It also suggests that higher temperature (lower $\beta$) reduces the bias. In this case, the logits get less weight, with the extreme of $\beta = 0$ corresponding to a uniform policy that has no bias. Lastly, the error scales linearly with $d$: the policy suffers from cumulative error as it relies on further-looking states in the approximate model.

## 5 SOFTTREEMAX: DEEP PARALLEL IMPLEMENTATION

Following impressive successes of deep RL (Mnih et al., 2015; Silver et al., 2016), using deep NNs in RL is standard practice. Depending on the RL algorithm, a loss function is defined and gradients on the network weights can be calculated. In PG methods, the scoring function used in the softmax is commonly replaced by a neural network $W_\theta$: $\pi_\theta(a|s) \propto \exp(W_\theta(s, a))$. Similarly, we implement SoftTreeMax by replacing $\theta(s)$ in (2) with a neural network $W_\theta(s)$. Although both variants of SoftTreeMax from Section 3 involve computing an expectation, this can be hard in general. One approach to handle it is with sampling, though these introduce estimation variance into the process. We leave the question of sample-based theory and algorithmic implementations for future work.

Instead, in finite action space environments such as Atari, we compute the exact expectation in SoftTreeMax with an exhaustive TS of depth $d$. Despite the exponential computational cost of spanning the entire tree, recent advancements in parallel GPU-based simulation allow efficient expansion of all nodes at the same depth simultaneously (Dalal et al., 2021; Rosenberg et al., 2022). This is possible when a simulator is implemented on GPU (Dalton et al., 2020; Makoviychuk et al., 2021; Freeman et al., 2021), or when a forward model is learned (Kim et al., 2020; Ha & Schmidhuber, 2018). To reduce the complexity to be linear in depth, we apply tree pruning to a limited width in all levels. We do so by sub-sampling only the most promising branches at each level. Limiting the width

drastically improves runtime, and enables respecting GPU memory limits, with only a small sacrifice in performance.

To summarize, in the practical SoftTreeMax algorithm we perform an exhaustive tree expansion with pruning to obtain trajectories up to depth $d$. We expand the tree with equal weight to all actions, which corresponds to a uniform tree expansion policy $\pi_b$. We apply a neural network on the leaf states, and accumulate the result with the rewards along each trajectory to obtain the logits in (2). Finally, we aggregate the results using C-SoftTreeMax. We leave experiments E-SoftTreeMax for future work on risk-averse RL. During training, the gradient propagates to the NN weights of $W_\theta$. When the gradient $\nabla_\theta \log \pi_{d,\theta}$ is calculated at each time step, it updates $W_\theta$ for all leaf states, similarly to Siamese networks (Bertinetto et al., 2016). An illustration of the policy is given in Figure 2.

## 6 EXPERIMENTS

We conduct our experiments on multiple games from the Atari simulation suite (Bellemare et al., 2013). As a baseline, we train a PPO (Schulman et al., 2017) agent with $256$ GPU workers in parallel (Dalton et al., 2020). For the tree expansion, we employ a GPU breadth-first as in (Dalal et al., 2021). We then train C-SoftTreeMax [1] for depths $d = 1 \ldots 8$, with a single worker. For depths $d \geq 3$, we limited the tree to a maximum width of $1024$ nodes and pruned trajectories with low estimated weights. Since the distributed PPO baseline advances significantly faster in terms of environment steps, for a fair comparison, we ran all experiments for one week on the same machine. For more details see Appendix B.

In Figure 3, we plot the reward and variance of SoftTreeMax for each game, as a function of depth. The dashed lines are the results for PPO. Each value is taken after convergence, i.e., the average over the last $20\%$ of the run. The numbers represent the average over five seeds per game. The plot conveys three intriguing conclusions. First, in all games, SoftTreeMax achieves significantly higher reward than PPO. Its gradient variance is also orders of magnitude lower than that of PPO. Second, the reward and variance are negatively correlated and mirror each other in almost all games. This phenomenon demonstrates the necessity of reducing the variance of PG for improving performance. Lastly, each game has a different sweet spot in terms of optimal tree depth. Recall that we limit the run-time in all experiments to one week The deeper the tree, the slower each step and the run consists of less steps. This explains the non-monotone behavior as a function of depth. For a more thorough discussion on the sweet spot of different games, see Appendix B.3.

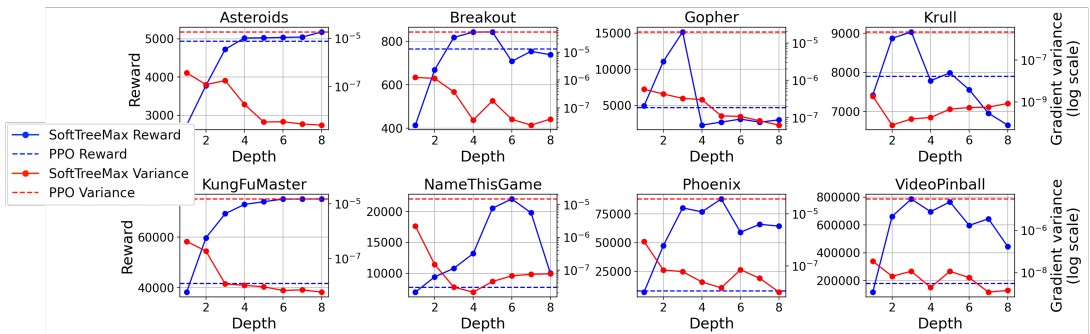

Figure 3: **Reward and Gradient variance: GPU SoftTreeMax (single worker) vs PPO (256 GPU workers).** The blue reward plots show the average of $50$ evaluation episodes. The red variance plots show the average gradient variance of the corresponding training runs, averaged over five seeds. The dashed lines represent the same for PPO. Note that the variance y-axis is in log-scale.

---

[1]We also experimented with E-SoftTreeMax and the results were almost identical. This is due to the quasi-deterministic nature of Atari, which causes the trajectory logits (2) to have almost no variability. We encourage future work on E-SoftTreeMax using probabilistic environments that are risk-sensitive.

## 7  RELATED WORK

**Softmax Operator.** The softmax policy became a canonical part of PG to the point where theoretical results of PG focus specifically on it (Zhang et al., 2021; Mei et al., 2020b; Li et al., 2021; Ding et al., 2022). Even though we focus on a tree extension to the softmax policy, our methodology is general and can be easily applied to other discrete or continuous parameterized policies as in (Mei et al., 2020a; Miahi et al., 2021; Silva et al., 2019). **Tree Search.** One famous TS algorithm is Monte-Carlo TS (MCTS; (Browne et al., 2012)) used in AlphaGo (Silver et al., 2016) and MuZero (Schrittwieser et al., 2020). Other algorithms such as Value Iteration, Policy Iteration and DQN were also shown to give an improved performance with a tree search extensions (Efroni et al., 2019; Dalal et al., 2021). **Parallel Environments.** In this work we used accurate parallel models that are becoming more common with the increasing popularity of GPU-based simulation (Makoviychuk et al., 2021; Dalton et al., 2020; Freeman et al., 2021). Alternatively, in relation to Theorem 4.8, one can rely on recent works that learn the underlying model (Ha & Schmidhuber, 2018; Schrittwieser et al., 2020) and use an approximation of the true dynamics. **Risk Aversion.** Previous work considered exponential utility functions for risk aversion (Chen et al., 2007; García & Fernández, 2015; Fei et al., 2021). This utility function is the same as E-SoftTreeMax formulation from (4), but we have it directly in the policy instead of the objective. **Reward-free RL.** We showed that the gradient variance is minimized when the transitions induced by the behavior policy $\pi_b$ are uniform. This is expressed by the second eigenvalue of the transition matrix $P^{\pi_b}$. This notion of uniform exploration is common to the reward-free RL setup (Jin et al., 2020). Several such works also considered the second eigenvalue in their analysis (Liu & Brunskill, 2018; Tarbouriech & Lazaric, 2019).

## 8  DISCUSSION

In this work, we introduced for the first time a differentiable parametric policy that combines TS with PG. We proved that SoftTreeMax is essentially a variance reduction technique and explained how to choose the expansion policy to minimize the gradient variance. It is an open question whether optimal variance reduction corresponds to the appealing regret properties the were put forward by UCT (Kocsis & Szepesvári, 2006). We believe that this can be answered by analyzing the convergence rate of SoftTreeMax, relying on the bias and variance results we obtained here.

As the learning process continues, the norm of the gradient and the variance *both* become smaller. On the face of it, one can ask if the gradient becomes small as fast as the variance or even faster can there be any meaningful learning? As we showed in the experiments, learning happens because the variance reduces fast enough (a variance of 0 represents deterministic learning, which is fastest).

Finally, our work can be extended to infinite action spaces. The analysis can be extended to infinite-dimension kernels that retain the same key properties used in our proofs. In the implementation, the tree of continuous actions can be expanded by maintaining a parametric distribution over actions that depend on $\theta$. This approach can be seen as a tree adaptation of MPPI (Williams et al., 2017).

### REPRODUCIBILITY AND LIMITATIONS

In this submission, we include the code as part of the supplementary material. We also include a docker file for setting up the environment and a README file with instructions on how to run both training and evaluation. The environment engine is an extension of Atari-CuLE (Dalton et al., 2020), a CUDA-based Atari emulator that runs on GPU. Our usage of a GPU environment is both a novelty and a current limitation of our work.

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

# APPENDIX

## A  PROOFS

### A.1  PROOF OF LEMMA 4.1 – BOUND ON THE POLICY GRADIENT VARIANCE

For any parametric policy $\pi_\theta$ and function $Q : \mathcal{S} \times \mathcal{A} \to \mathbb{R}$,

$$\text{Var}\left(\nabla_\theta \log \pi_\theta(a|s) Q(s,a)\right) \leq \max_{s,a} \left[Q(s,a)\right]^2 \max_s \|\nabla_\theta \log \pi_\theta(\cdot|s)\|_F^2,$$

where $\nabla_\theta \log \pi_\theta(\cdot|s) \in \mathbb{R}^{A \times \dim(\theta)}$ is a matrix whose $a$-th row is $\nabla_\theta \log \pi_\theta(a|s)^\top$.

*Proof.* The variance for a parametric policy $\pi_\theta$ is given as follows:

$$\text{Var}\left(\nabla_\theta \log \pi_\theta(a|s)Q(a,s)\right) = \mathbb{E}_{s \sim d_{\pi_\theta}, a \sim \pi_\theta(\cdot|s)}\left[\nabla_\theta \log \pi_\theta(a|s)^\top \nabla_\theta \log \pi_\theta(a|s)Q(s,a)^2\right] -$$
$$\mathbb{E}_{s \sim d_{\pi_\theta}, a \sim \pi_\theta(\cdot|s)}\left[\nabla_\theta \log \pi_\theta(a|s)Q(s,a)\right]^\top \mathbb{E}_{s \sim d_{\pi_\theta}, a \sim \pi_\theta(\cdot|s)}\left[\nabla_\theta \log \pi_\theta(a|s)Q(s,a)\right],$$

where $Q(s,a)$ is the currently estimated Q-function and $d_{\pi_\theta}$ is the discounted state visitation frequency induced by the policy $\pi_\theta$. Since the second term we subtract is always positive (it is of quadratic form $v^\top v$) we can bound the variance by the first term:

$$\text{Var}\left(\nabla_\theta \log \pi_\theta(a|s)Q(a,s)\right) \le \mathbb{E}_{s \sim d_{\pi_\theta}, a \sim \pi_\theta(\cdot|s)}\left[\nabla_\theta \log \pi_\theta(a|s)^\top \nabla_\theta \log \pi_\theta(a|s)Q(s,a)^2\right]$$
$$= \sum_s d_{\pi_\theta}(s) \sum_a \pi_\theta(a|s)\nabla_\theta \log \pi_\theta(a|s)^\top \nabla_\theta \log \pi_\theta(a|s)Q(s,a)^2$$
$$\le \max_{s,a}\left[[Q(s,a)]^2 \pi_\theta(a|s)\right] \sum_s d_{\pi_\theta}(s) \sum_a \nabla_\theta \log \pi_\theta(a|s)^\top \nabla_\theta \log \pi_\theta(a|s)$$
$$\le \max_{s,a}[Q(s,a)]^2 \max_s \sum_a \nabla_\theta \log \pi_\theta(a|s)^\top \nabla_\theta \log \pi_\theta(a|s)$$
$$= \max_{s,a}[Q(s,a)]^2 \max_s \|\nabla_\theta \log \pi_\theta(\cdot|s)\|_F^2.$$

$\square$

### A.2 PROOF OF LEMMA 4.2 – VECTOR FORM OF C-SOFTTREEMAX

In vector form, (3) is given by

$$\pi_{d,\theta}^{\text{C}}(\cdot|s) = \frac{\exp\left[\beta\left(C_{s,d} + P_s\left(P^{\pi_b}\right)^{d-1}\Theta\right)\right]}{\mathbf{1}_A^\top \exp\left[\beta\left(C_{s,d} + P_s\left(P^{\pi_b}\right)^{d-1}\Theta\right)\right]}, \tag{8}$$

where

$$C_{s,d} = \gamma^{-d}R_s + P_s\left[\sum_{h=1}^{d-1}\gamma^{h-d}\left(P^{\pi_b}\right)^{h-1}\right]R_{\pi_b}. \tag{9}$$

*Proof.* Consider the vector $\ell_{s,\cdot} \in \mathbb{R}^{|\mathcal{A}|}$. Its expectation satisfies

$$\mathbb{E}^{\pi_b}\ell_{s,\cdot}(d;\theta) = \mathbb{E}^{\pi_b}\left[\sum_{t=0}^{d-1}\gamma^{t-d}r_t + \theta(s_d)\right]$$
$$= \gamma^{-d}R_s + \sum_{t=1}^{d-1}\gamma^{t-d}P_s(P^{\pi_b})^{t-1}R_{\pi_b} + P_s(P^{\pi_b})^{d-1}\Theta.$$

As required. $\square$

### A.3 PROOF OF LEMMA 4.3 – GRADIENT OF C-SOFTTREEMAX

The C-SoftTreeMax gradient of dimension $A \times S$ is given by

$$\nabla_\theta \log \pi_{d,\theta}^{\text{C}} = \beta\left[I_A - \mathbf{1}_A(\pi_{d,\theta}^{\text{C}})^\top\right]P_s\left(P^{\pi_b}\right)^{d-1},$$

where for brevity, we drop the $s$ index in the policy above, i.e., $\pi_{d,\theta}^{\text{C}} \equiv \pi_{d,\theta}^{\text{C}}(\cdot|s)$.

*Proof.* The $(j,k)$-th entry of $\nabla_\theta \log \pi_{d,\theta}^{\mathsf{C}}$ satisifes

$$[\nabla_\theta \log \pi_{d,\theta}^{\mathsf{C}}]_{j,k} = \frac{\partial \log(\pi_{d,\theta}^{\mathsf{C}}(a^j|s))}{\partial \theta(s^k)}$$

$$= \beta[P_s(P^{\pi_b})^{d-1}]_{j,k} - \frac{\sum_a \left[\exp\left[\beta\left(C_{s,d} + P_s\left(P^{\pi_b}\right)^{d-1}\Theta\right)\right]\right]_a \beta\left[P_s(P^{\pi_b})^{d-1}\right]_{a,k}}{\mathbf{1}_A^\top \exp\left[\beta\left(C_{s,d} + P_s\left(P^{\pi_b}\right)^{d-1}\Theta\right)\right]}$$

$$= \beta[P_s(P^{\pi_b})^{d-1}]_{j,k} - \beta\sum_a \pi_{d,\theta}^{\mathsf{C}}(a|s)\left[P_s(P^{\pi_b})^{d-1}\right]_{a,k}$$

$$= \beta[P_s(P^{\pi_b})^{d-1}]_{j,k} - \beta\left[(\pi_{d,\theta}^{\mathsf{C}})^\top P_s(P^{\pi_b})^{d-1}\right]_k$$

$$= \beta[P_s(P^{\pi_b})^{d-1}]_{j,k} - \beta\left[\mathbf{1}_A(\pi_{d,\theta}^{\mathsf{C}})^\top P_s(P^{\pi_b})^{d-1}\right]_{j,k}.$$

Moving back to matrix form, we obtain the stated result. $\qquad\square$

### A.4 PROOF OF THEOREM 4.4 – EXPONENTIAL VARIANCE DECAY OF C-SOFTTREEMAX

The C-SoftTreeMax policy gradient is bounded by

$$\mathrm{Var}\left(\nabla_\theta \log \pi_{d,\theta}^{\mathsf{C}}(a|s)Q(s,a)\right) \leq 2\frac{A^2 S^2 \beta^2}{(1-\gamma)^2}|\lambda_2(P^{\pi_b})|^{2(d-1)}.$$

*Proof.* We use Lemma 4.1 directly. First of all, it is know that when the reward is bounded in $[0,1]$, the maximal value of the Q-function is $\frac{1}{1-\gamma}$ as the sum as infinite discounted rewards. Next, we bound the Frobenius norm of the term achieved in Lemma 4.3, by applying the eigen-decomposition on $P^{\pi_b}$:

$$P^{\pi_b} = \mathbf{1}_S \mu^\top + \sum_{i=2}^{S} \lambda_i u_i v_i^\top, \tag{10}$$

where $\mu$ is the stationary distribution of $P^{\pi_b}$, and $u_i$ and $v_i$ are left and right eigenvectors correspondingly.

$$\|\beta\left(I_{A,A} - \mathbf{1}_A\pi^\top\right)P_s(P^{\pi_b})^{d-1}\|_F = \beta\|\left(I_{A,A} - \mathbf{1}_A\pi^\top\right)P_s\left(\mathbf{1}_S\mu^\top + \sum_{i=2}^{S}\lambda_i^{d-1}u_i v_i^\top\right)\|_F$$

$$(P_s \text{ is stochastic}) \quad = \beta\|\left(I_{A,A} - \mathbf{1}_A\pi^\top\right)\left(\mathbf{1}_A\mu^\top + \sum_{i=2}^{S}\lambda_i^{d-1}P_s u_i v_i^\top\right)\|_F$$

$$(\text{projection nullifies } \mathbf{1}_A\mu^\top) \quad = \beta\|\left(I_{A,A} - \mathbf{1}_A\pi^\top\right)\left(\sum_{i=2}^{S}\lambda_i^{d-1}P_s u_i v_i^\top\right)\|_F$$

$$(\text{triangle inequality}) \quad \leq \beta\sum_{i=2}^{S}\|\left(I_{A,A} - \mathbf{1}_A\pi^\top\right)\left(\lambda_i^{d-1}P_s u_i v_i^\top\right)\|_F$$

$$(\text{matrix norm sub-multiplicativity}) \quad \leq \beta|\lambda_2^{d-1}|\sum_{i=2}^{S}\|I_{A,A} - \mathbf{1}_A\pi^\top\|_F\|P_s\|_F\|u_i v_i^\top\|_F$$

$$= \beta|\lambda_2^{d-1}|(S-1)\|I_{A,A} - \mathbf{1}_A\pi^\top\|_F\|P_s\|_F.$$

Now, we can bound the norm $\|I_{A,A} - \mathbf{1}_A\pi^\top\|_F$ by direct calculation:

$$\|I_{A,A} - \mathbf{1}_A\pi^\top\|_F^2 = \mathrm{Tr}\left[\left(I_{A,A} - \mathbf{1}_A\pi^\top\right)\left(I_{A,A} - \mathbf{1}_A\pi^\top\right)^\top\right] \tag{11}$$

$$= \mathrm{Tr}\left[I_{A,A} - \mathbf{1}_A\pi^\top - \pi\mathbf{1}_A^\top + \pi^\top\pi\mathbf{1}_A\mathbf{1}_A^\top\right] \tag{12}$$

$$= A - 1 - 1 + A\pi^\top\pi \tag{13}$$

$$\leq 2A. \tag{14}$$

From the Cauchy-Schwartz inequality,

$$\|P_s\|_F^2 = \sum_a \sum_s [[P_s]_{a,s}]^2 = \sum_a \|[P_s]_{a,\cdot}\|_2^2 \leq \sum_a \|[P_s]_{a,\cdot}\|_1 \|[P_s]_{a,\cdot}\|_\infty \leq A.$$

So,

$$\text{Var}\left(\nabla_\theta \log \pi_{d,\theta}^{\mathrm{C}}(a|s)Q(s,a)\right) \leq \max_{s,a}[Q(s,a)]^2 \max_s \|\nabla_\theta \log \pi_{d,\theta}^{\mathrm{C}}(\cdot|s)\|_F^2$$

$$\leq \frac{1}{(1-\gamma)^2}\|\beta\left(I_{A,A} - \mathbf{1}_A\pi^\top\right)P_s(P^{\pi_b})^{d-1}\|_F^2$$

$$\leq \frac{1}{(1-\gamma)^2}\beta^2|\lambda_2(P^{\pi_b})|^{2(d-1)}S^2(2A^2),$$

which obtains the desired bound. $\qquad\square$

### A.5 A LOWER BOUND ON C-SOFTTREEMAX GRADIENT (RESULT NOT IN THE PAPER)

For completeness we also supply a lower bound on the Frobenius norm of the gradient. Note that this result does not translate to the a lower bound on the variance since we have no lower bound equivalence of Lemma 4.1.

**Lemma A.1.** *The Frobenius norm on the gradient of the policy is lower-bounded by:*

$$\|\nabla_\theta \log \pi_{d,\theta}^C(\cdot|s)\|_F \geq C \cdot \beta|\lambda_2(P^{\pi_b})|^{(d-1)}. \tag{15}$$

*Proof.* We begin by moving to the induced $l_2$ norm by norm-equivalence:

$$\|\beta\left(I_{A,A} - \mathbf{1}_A\pi^\top\right)P_s(P^{\pi_b})^{d-1}\|_F \geq \|\beta\left(I_{A,A} - \mathbf{1}_A\pi^\top\right)P_s(P^{\pi_b})^{d-1}\|_2.$$

Now, taking the vector $u$ to be the eigenvector of the second eigenvalue of $P^{\pi_b}$:

$$\|\beta\left(I_{A,A} - \mathbf{1}_A\pi^\top\right)P_s(P^{\pi_b})^{d-1}\|_2 \geq \|\beta\left(I_{A,A} - \mathbf{1}_A\pi^\top\right)P_s(P^{\pi_b})^{d-1}u\|_2$$

$$= \beta\|\left(I_{A,A} - \mathbf{1}_A\pi^\top\right)P_su\|_2$$

$$= \beta|\lambda_2(P^{\pi_b})|^{(d-1)}\|\left(I_{A,A} - \mathbf{1}_A\pi^\top\right)P_su\|_2.$$

Note that even though $P_su$ can be 0, that is not the common case since we can freely change $\pi_b$ (and therefore the eigenvectors of $P^{\pi_b}$). $\qquad\square$

### A.6 PROOF OF LEMMA 4.5 – VECTOR FORM OF E-SOFTTREEMAX

For $d \geq 1$, (4) is given by

$$\pi_{d,\theta}^{\mathrm{E}}(\cdot|s) = \frac{E_{s,d}\exp(\beta\Theta)}{\mathbf{1}_A^\top E_{s,d}\exp(\beta\Theta)}, \tag{16}$$

where

$$E_{s,d} = P_s \prod_{h=1}^{d-1}\left(D\left(\exp[\beta\gamma^{h-d}R]\right)P^{\pi_b}\right) \tag{17}$$

with $R$ being the $|S|$-dimensional vector whose $s$-th coordinate is $r(s)$.

*Proof.* Recall that

$$\ell_{s,a}(d;\theta) = \gamma^{-d}\left[r(s) + \sum_{t=1}^{d-1}\gamma^t r(s_t) + \gamma^d\theta(s_d)\right]. \tag{18}$$

and, hence,

$$\exp[\beta\ell_{s,a}(d;\theta)] = \exp\left[\beta\gamma^{-d}\left(r(s) + \sum_{t=1}^{d-1}\gamma^t r(s_t) + \gamma^d\theta(s_d)\right)\right]. \tag{19}$$

Therefore,

$$\mathbb{E}[\exp \beta \ell_{s,a}(d;\theta)] = \mathbb{E}\left[\exp\left[\beta\gamma^{-d}\left(r(s) + \sum_{t=1}^{d-1}\gamma^t r(s_t)\right)\right]\mathbb{E}\left[\exp\left[\beta\left(\theta(s_d)\right)\right]|s_1,\ldots,s_{d-1}\right]\right] \tag{20}$$

$$= \mathbb{E}\left[\exp\left[\beta\gamma^{-d}\left(r(s) + \sum_{t=1}^{d-1}\gamma^t r(s_t)\right)\right]P^{\pi_b}(\cdot|s_{d-1})\exp(\beta\Theta)\right] \tag{21}$$

$$= \mathbb{E}\left[\exp\left[\beta\gamma^{-d}\left(r(s) + \sum_{t=1}^{d-2}\gamma^t r(s_t)\right)\right]\exp[\beta\gamma^{-1}r(s_{d-1})]P^{\pi_b}(\cdot|s_{d-1})\right]\exp(\beta\Theta). \tag{22}$$

By repeatedly using iterative conditioning as above, the desired result follows. Note that $\exp(\beta\gamma^{-d}r(s))$ does not depend on the action and is therefore cancelled out with the denominator. $\square$

## A.7 PROOF OF LEMMA 4.6 – GRADIENT OF E-SOFTTREEMAX

The E-SoftTreeMax gradient of dimension $A \times S$ is given by

$$\nabla_\theta \log \pi_{d,\theta}^{\mathrm{E}} = \beta \left[I_A - \mathbf{1}_A(\pi_{d,\theta}^{\mathrm{E}})^\top\right] \frac{D\left(\pi_{d,\theta}^{\mathrm{E}}\right)^{-1} E_{s,d} D(\exp(\beta\Theta))}{\mathbf{1}_A^\top E_{s,d}\exp(\beta\Theta)},$$

where for brevity, we drop the $s$ index in the policy above, i.e., $\pi_{d,\theta}^{\mathrm{E}} \equiv \pi_{d,\theta}^{\mathrm{E}}(\cdot|s)$.

*Proof.* The $(j,k)$-th entry of $\nabla_\theta \log \pi_{d,\theta}^{\mathrm{E}}$ satisfies

$$[\nabla_\theta \log \pi_{d,\theta}^{\mathrm{E}}]_{j,k} = \frac{\partial \log(\pi_{d,\theta}^{\mathrm{E}}(a^j|s))}{\partial\theta(s^k)}$$

$$= \frac{\partial}{\partial\theta(s^k)}\left(\log[(E_{s,d})_j^\top \exp(\beta\Theta)] - \log[\mathbf{1}_A^\top E_{s,d}\exp(\beta\Theta)]\right)$$

$$= \frac{\beta(E_{s,d})_{j,k}\exp(\beta\theta(s^k))}{(E_{s,d})_j^\top \exp(\beta\Theta)} - \frac{\beta\mathbf{1}_A^\top E_{s,d}e_k\exp(\beta\theta(s^k))}{\mathbf{1}_A^\top E_{s,d}\exp(\beta\Theta)}$$

$$= \frac{\beta(E_{s,d}e_k\exp(\beta\theta(s^k)))_j}{(E_{s,d})_j^\top \exp(\beta\Theta)} - \frac{\beta\mathbf{1}_A^\top E_{s,d}e_k\exp(\beta\theta(s^k))}{\mathbf{1}_A^\top E_{s,d}\exp(\beta\Theta)}$$

$$= \beta\left[\frac{e_j^\top}{e_j^\top E_{s,d}\exp(\beta\Theta)} - \frac{\mathbf{1}_A^\top}{\mathbf{1}_A^\top E_{s,d}\exp(\beta\Theta)}\right]E_{s,d}e_k\exp(\beta\theta(s^k)).$$

Hence,

$$[\nabla_\theta \log \pi_{d,\theta}^{\mathrm{E}}]_{\cdot,k} = \beta\left[D(E_{s,d}\exp(\beta\Theta))^{-1} - (\mathbf{1}_A^\top E_{s,d}\exp(\beta\Theta))^{-1}\mathbf{1}_A\mathbf{1}_A^\top\right]E_{s,d}e_k\exp(\beta\theta(s^k))$$

From this, it follows that

$$\nabla_\theta \log \pi_{d,\theta}^{\mathrm{E}} = \beta\left[D\left(\pi_{d,\theta}^{\mathrm{E}}\right)^{-1} - \mathbf{1}_A\mathbf{1}_A^\top\right]\frac{E_{s,d}D(\exp(\beta\Theta))}{\mathbf{1}_A^\top E_{s,d}\exp(\beta\Theta)}. \tag{23}$$

The desired result is now easy to see. $\square$

## A.8 PROOF OF THEOREM 4.7 — EXPONENTIAL VARIANCE DECAY OF E-SOFTTREEMAX

There exists $\alpha \in (0,1)$ such that, for any function $Q : \mathcal{S} \times \mathcal{A} \to \mathbb{R}$,

$$\mathrm{Var}\left(\nabla_\theta \log \pi_{d,\theta}^{\mathrm{E}}(a|s)Q(s,a)\right) \in \mathcal{O}\left(\beta^2\alpha^{2d}\right).$$

If all rewards are equal ($r \equiv$ const), then $\alpha = |\lambda_2(P^{\pi_b})|$.

*Proof outline.* Recall that thanks to Lemma 4.1, we can bound the PG variance using a direct bound on the gradient norm. The definition of the induced norm is

$$\|\nabla_\theta \log \pi_{d,\theta}^{\mathrm{E}}\| = \max_{z:\|z\|=1} \|\nabla_\theta \log \pi_{d,\theta}^{\mathrm{E}} z\|,$$

with $\nabla_\theta \log \pi_{d,\theta}^{\mathrm{E}}$ given in Lemma 4.6. Let $z \in \mathbb{R}^S$ be an arbitrary vector such that $\|z\| = 1$. Then, $z = \sum_{i=1}^S c_i z_i$, where $c_i$ are scalar coefficients and $z_i$ are vectors spanning the $S$-dimensional space. In the full proof, we show our specific choice of $z_i$ and prove they are linearly independent given that choice. We do note that $z_1 = \mathbf{1}_S$.

The first part of the proof relies on the fact that $(\nabla_\theta \log \pi_{d,\theta}^{\mathrm{E}}) z_1 = 0$. This is easy to verify using Lemma 4.6 together with (6), and because $\left[ I_A - \mathbf{1}_A (\pi_{d,\theta}^{\mathrm{E}})^\top \right]$ is a projection matrix whose null-space is spanned by $\mathbf{1}_S$. Thus,

$$\nabla_\theta \log \pi_{d,\theta}^{\mathrm{E}} z = \nabla_\theta \log \pi_{d,\theta}^{\mathrm{E}} \sum_{i=2}^S c_i z_i.$$

In the second part of the proof, we focus on $E_{s,d}$ from (6), which appears within $\nabla_\theta \log \pi_{d,\theta}^{\mathrm{E}}$. Notice that $E_{s,d}$ consists of the product $\prod_{h=1}^{d-1} \left( D \left( \exp(\beta \gamma^{h-d} R) \right) P^{\pi_b} \right)$. Even though the elements in this product are not stochastic matrices, in the full proof we show how to normalize each of them to a stochastic matrix $B_h$. We thus obtain that

$$E_{s,d} = P_s D(M_1) \prod_{h=1}^{d-1} B_h,$$

where $M_1 \in \mathbb{R}^S$ is some strictly positive vector. Then, we can apply a result by Mathkar & Borkar (2016), which itself builds on (Chatterjee & Seneta, 1977). The result states that the product of stochastic matrices $\prod_{h=1}^{d-1} B_h$ of our particular form converges exponentially fast to a matrix of the form $\mathbf{1}_S \mu^\top$ s.t. $\|\mathbf{1}_S \mu^\top - \prod_{h=1}^{d-1} B_h\| \leq C\alpha^d$ for some constant $C$.

Lastly, $\mathbf{1}_S \mu_{\pi_b}^\top$ gets canceled due to our choice of $z_i$, $i = 2, \ldots, S$. This observation along with the above fact that the remainder decays then shows that $\nabla_\theta \log \pi_{d,\theta}^{\mathrm{E}} \sum_{i=2}^S z_i = \mathcal{O}(\alpha^d)$, which gives the desired result. $\qquad\square$

*Full technical proof.* Let $d \geq 2$. Recall that

$$E_{s,d} = P_s \prod_{h=1}^{d-1} \left( D \left( \exp[\beta \gamma^{h-d} R] \right) P^{\pi_b} \right), \tag{24}$$

and that $R$ refers to the $S$-dimensional vector whose $s$-th coordinate is $r(s)$. Define

$$B_i = \begin{cases} P^{\pi_b} & \text{if } i = d-1, \\ D^{-1}(P^{\pi_b} M_{i+1}) P^{\pi_b} D(M_{i+1}) & \text{if } i = 1, \ldots, d-2, \end{cases} \tag{25}$$

and the vector

$$M_i = \begin{cases} \exp(\beta \gamma^{-1} R) & \text{if } i = d-1, \\ \exp(\beta \gamma^{i-d} R) \circ P^{\pi_b} M_{i+1} & \text{if } i = 1, \ldots, d-2, \end{cases} \tag{26}$$

where $\circ$ denotes the element-wise product. Then,

$$E_{s,d} = P_s D(M_1) \prod_{i=1}^{d-1} B_i. \tag{27}$$

It is easy to see that each $B_i$ is a row-stochastic matrix, i.e., all entries are non-negative and $B_i \mathbf{1}_S = \mathbf{1}_S$.

Next, we prove that all non-zeros entries of $B_i$ are bounded away from 0 by a constant. This is necessary to apply the next result from Chatterjee & Seneta (1977). The $j$-th coordinate of $M_i$ satisfies

$$(M_i)_j = \exp[\beta \gamma^{i-d} R_j] \sum_k [P^{\pi_b}]_{j,k} (M_{i+1})_k \leq \| \exp[\beta \gamma^{i-d} R] \|_\infty \| M_{i+1} \|_\infty. \tag{28}$$

Separately, observe that $\|M_{d-1}\|_\infty \leq \|\exp(\beta\gamma^{-1}R)\|_\infty$. Plugging these relations in (26) gives

$$\|M_1\|_\infty \leq \prod_{h=1}^{d-1} \|\exp[\beta\gamma^{h-d}R]\|_\infty = \prod_{h=1}^{d-1} \|\exp[\beta\gamma^{-d}R]\|_\infty^{\gamma^h} = \|\exp[\beta\gamma^{-d}R]\|_\infty^{\sum_{h=1}^{d-1}\gamma^h} \leq \|\exp[\beta\gamma^{-d}R]\|_\infty^{\frac{1}{1-\gamma}}. \tag{29}$$

Similarly, for every $1 \leq i \leq d-1$, we have that

$$\|M_i\|_\infty \leq \prod_{h=i}^{d-1} \|\exp[\beta\gamma^{-d}R]\|_\infty^{\gamma^h} \leq \|\exp[\beta\gamma^{-d}R]\|_\infty^{\frac{1}{1-\gamma}}. \tag{30}$$

The $jk$-th entry of $B_i = D^{-1}(P^{\pi_b}M_{i+1})P^{\pi_b}D(M_{i+1})$ is

$$(B_i)_{jk} = \frac{P_{jk}^{\pi_b}[M_{i+1}]_k}{\sum_{\ell=1}^{|S|} P_{j\ell}^{\pi_b}[M_{i+1}]_\ell} \geq \frac{P_{jk}^{\pi_b}}{\sum_{\ell=1}^{|S|} P_{j\ell}^{\pi_b}[M_{i+1}]_\ell} \geq \frac{P_{jk}^{\pi_b}}{\|\exp[\beta\gamma^{-d}R]\|_\infty^{\frac{1}{1-\gamma}}}. \tag{31}$$

Hence, for non-zero $P_{jk}^{\pi_b}$, the entries are bounded away from zero by the same. We can now proceed with applying the following result.

Now, by (Chatterjee & Seneta, 1977, Theorem 5) (see also (14) in (Mathkar & Borkar, 2016)), $\lim_{d\to\infty} \prod_{i=1}^{d-1} B_i$ exists and is of the form $\mathbf{1}_S\mu^\top$ for some probability vector $\mu$. Furthermore, there is some $\alpha \in (0,1)$ such that $\varepsilon(d) := \left(\prod_{i=1}^{d-1} B_i\right) - \mathbf{1}_S\mu^\top$ satisfies

$$\|\varepsilon(d)\| = O(\alpha^d). \tag{32}$$

Pick linearly independent vectors $w_2, \ldots, w_S$ such that

$$\mu^\top w_i = 0 \text{ for } i = 2, \ldots, d. \tag{33}$$

Since $\sum_{i=2}^S \alpha_i w_i$ is perpendicular to $\mu$ for any $\alpha_2, \ldots \alpha_S$ and because $\mu^\top \exp(\beta\Theta) > 0$, there exists no choice of $\alpha_2, \ldots, \alpha_S$ such that $\sum_{i=2}^S \alpha_i w_i = \exp(\beta\Theta)$. Hence, if we let $z_1 = \mathbf{1}_S$ and $z_i = D(\exp(\beta\Theta))^{-1}w_i$ for $i = 2, \ldots, S$, then it follows that $\{z_1, \ldots, z_S\}$ is linearly independent. In particular, it implies that $\{z_1, \ldots, z_S\}$ spans $\mathbb{R}^S$.

Now consider an arbitrary unit norm vector $z := \sum_{i=1}^S c_i z_i \in \mathbb{R}^S$ s.t. $\|z\|_2 = 1$. Then,

$$\nabla_\theta \log \pi_{d,\theta}^{\mathrm{E}} z = \nabla_\theta \log \pi_{d,\theta}^{\mathrm{E}} \sum_{i=2}^S c_i z_i \tag{34}$$

$$= \beta \left[I_A - \mathbf{1}_A(\pi_{d,\theta}^{\mathrm{E}})^\top\right] \frac{D\left(\pi_{d,\theta}^{\mathrm{E}}\right)^{-1} E_{s,d} D(\exp(\beta\Theta))}{\mathbf{1}_A^\top E_{s,d} \exp(\beta\Theta)} \sum_{i=2}^S c_i z_i \tag{35}$$

$$= \beta \left[I_A - \mathbf{1}_A(\pi_{d,\theta}^{\mathrm{E}})^\top\right] \frac{D\left(\pi_{d,\theta}^{\mathrm{E}}\right)^{-1} E_{s,d}}{\mathbf{1}_A^\top E_{s,d} \exp(\beta\Theta)} \sum_{i=2}^S c_i w_i \tag{36}$$

$$= \beta \left[I_A - \mathbf{1}_A(\pi_{d,\theta}^{\mathrm{E}})^\top\right] \frac{D\left(\pi_{d,\theta}^{\mathrm{E}}\right)^{-1} \left[\mathbf{1}_S\mu^\top + \varepsilon(d)\right]}{\mathbf{1}_A^\top E_{s,d} \exp(\beta\Theta)} \sum_{i=2}^S c_i w_i \tag{37}$$

$$= \beta \left[I_A - \mathbf{1}_A(\pi_{d,\theta}^{\mathrm{E}})^\top\right] \frac{D\left(\pi_{d,\theta}^{\mathrm{E}}\right)^{-1} \varepsilon(d)}{\mathbf{1}_A^\top E_{s,d} \exp(\beta\Theta)} \sum_{i=2}^S c_i w_i \tag{38}$$

$$= \beta \left[I_A - \mathbf{1}_A(\pi_{d,\theta}^{\mathrm{E}})^\top\right] \frac{D\left(\pi_{d,\theta}^{\mathrm{E}}\right)^{-1} \varepsilon(d) D(\exp(\beta\Theta))}{\mathbf{1}_A^\top E_{s,d} \exp(\beta\Theta)} (z - c_1\mathbf{1}_S), \tag{39}$$

where (34) follows from the fact that $\nabla_\theta \log \pi_{d,\theta}^{\mathrm{E}} z_1 = \nabla_\theta \log \pi_{d,\theta}^{\mathrm{E}} \mathbf{1}_S = 0$, (35) follows from Lemma 4.6, (36) holds since $z_i = D(\exp(\beta\Theta))^{-1} w_i$, (38) because $\mu$ is perpendicular $w_i$ for each $i$, while (39) follows by reusing $z_i = D(\exp(\beta\Theta))^{-1} w_i$ relation along with the fact that $z_1 = \mathbf{1}_S$.

From (39), it follows that

$$\|\nabla_\theta \log \pi_{d,\theta}^{\mathrm{E}} z\| \leq \beta\|\varepsilon(d)\| \left\| \left[I_A - \mathbf{1}_A(\pi_{d,\theta}^{\mathrm{E}})^\top\right] \frac{D\left(\pi_{d,\theta}^{\mathrm{E}}\right)^{-1}}{\mathbf{1}_A^\top E_{s,d} \exp(\beta\Theta)} \right\| \|D(\exp(\beta\Theta))\| \|z - c_1 \mathbf{1}_S\| \tag{40}$$

$$\leq \beta\alpha^d (\|I_A\| + \|\mathbf{1}_A(\pi_{d,\theta}^{\mathrm{E}})^\top\|) \left\| \frac{D\left(\pi_{d,\theta}^{\mathrm{E}}\right)^{-1}}{\mathbf{1}_A^\top E_{s,d} \exp(\beta\Theta)} \right\| \exp(\beta \max_s \theta(s)) \|z - c_1 \mathbf{1}_S\| \tag{41}$$

$$\leq \beta\alpha^d (1 + \sqrt{A}) \left\| \frac{D\left(\pi_{d,\theta}^{\mathrm{E}}\right)^{-1}}{\mathbf{1}_A^\top E_{s,d} \exp(\beta\Theta)} \right\| \exp(\beta \max_s \theta(s)) \|z - c_1 \mathbf{1}_S\| \tag{42}$$

$$\leq \beta\alpha^d (1 + \sqrt{A}) \left\| D^{-1}(E_{s,d} \exp(\beta\Theta)) \right\| \exp(\beta \max_s \theta(s)) \|z - c_1 \mathbf{1}_S\| \tag{43}$$

$$\leq \beta\alpha^d (1 + \sqrt{A}) \frac{1}{\min_s [E_{s,d} \exp(\beta\Theta)]_s} \exp(\beta \max_s \theta(s)) \|z - c_1 \mathbf{1}_S\| \tag{44}$$

$$\leq \beta\alpha^d (1 + \sqrt{A}) \frac{\exp(\beta \max_s \theta(s))}{\exp(\beta \min_s \theta(s)) \min_s |M_1|} \|z - c_1 \mathbf{1}_S\| \tag{45}$$

$$\leq \beta\alpha^d (1 + \sqrt{A}) \frac{\exp(\beta \max_s \theta(s))}{\exp(\beta \min_s \theta(s)) \exp(\beta \min_s r(s))} \|z - c_1 \mathbf{1}_S\| \tag{46}$$

$$\leq \beta\alpha^d (1 + \sqrt{A}) \exp(\beta[\max_s \theta(s) - \min_s \theta(s) - \min_s r(s)]) \|z - c_1 \mathbf{1}_S\|. \tag{47}$$

Lastly, we prove that $\|z - c_1 \mathbf{1}_S\|$ is bounded independently of $d$. First, denote by $c = (c_1, \ldots, c_S)^\top$ and $\tilde{c} = (0, c_2, \ldots, c_S)^\top$. Also, denote by $Z$ the matrix with $z_i$ as its $i$-th column. Now,

$$\|z - c_1 \mathbf{1}_S\| = \|\sum_{i=2}^{S} c_i z_i\| \tag{48}$$

$$= \|Z\tilde{c}\| \tag{49}$$

$$\leq \|Z\|\|\tilde{c}\| \tag{50}$$

$$\leq \|Z\|\|c\| \tag{51}$$

$$= \|Z\|\|Z^{-1} z\| \tag{52}$$

$$\leq \|Z\|\|Z^{-1}\|, \tag{53}$$

where the last relation is due to $z$ being a unit vector. All matrix norms here are $l_2$-induced norms.

Next, denote by $W$ the matrix with $w_i$ in its $i$-th column. Recall that in (33) we only defined $w_2, \ldots, w_S$. We now set $w_1 = \exp(\beta\Theta)$. Note that $w_1$ is linearly independent of $\{w_2, \ldots, w_S\}$ because of (33) together with the fact that $\mu^\top w_1 > 0$. We can now express the relation between $Z$ and $W$ by $Z = D^{-1}(\exp(\beta\Theta))W$. Substituting this in (53), we have

$$\|z - c_1 \mathbf{1}_S\| \leq \|D^{-1}(\exp(\beta\Theta))W\|\|W^{-1} D(\exp(\beta\Theta))\| \tag{54}$$

$$\leq \|W\|\|W^{-1}\|\|D(\exp(\beta\Theta))\|\|D^{-1}(\exp(\beta\Theta))\|. \tag{55}$$

It further holds that

$$\|D(\exp(\beta\Theta))\| \leq \max_s \exp(\beta\theta(s)) \leq \max\{1, \exp[\beta \max_s \theta(s)]\}, \tag{56}$$

where the last relation equals 1 if $\theta(s) < 0$ for all $s$. Similarly,

$$\|D^{-1}(\exp(\beta\Theta))\| \leq \frac{1}{\min_s \exp(\beta\theta(s))} \leq \frac{1}{\min\{1, \exp[\beta \min_s \theta(s)]\}}. \tag{57}$$

Furthermore, by the properties of the $l_2$-induced norm,

$$\|W\|_2 \leq \sqrt{S}\|W\|_1 \tag{58}$$

$$= \sqrt{S} \max_{1 \leq i \leq S} \|w_i\|_1 \tag{59}$$

$$= \sqrt{S} \max\{\exp(\beta\Theta), \max_{2 \leq i \leq S} \|w_i\|_1\} \tag{60}$$

$$\leq \sqrt{S} \max\{1, \exp[\beta \max_s \theta(s)], \max_{2 \leq i \leq S} \|w_i\|_1)\}. \tag{61}$$

Lastly,

$$\|W^{-1}\| = \frac{1}{\sigma_{\min}(W)} \tag{62}$$

$$\leq \left(\prod_{i=1}^{S-1} \frac{\sigma_{\max}(W)}{\sigma_i(W)}\right) \frac{1}{\sigma_{\min}(W)} \tag{63}$$

$$= \frac{(\sigma_{\max}(W))^{S-1}}{\prod_{i=1}^S \sigma_i(W)} \tag{64}$$

$$= \frac{\|W\|^{S-1}}{|\det(W)|}. \tag{65}$$

The determinant of $W$ is a sum of products involving its entries. To upper bound (65) independently of $d$, we lower bound its denominator by upper and lower bounds on the entries $[W]_{i,1}$ that are independent of $d$, depending on their sign:

$$\min\{1, \exp[\beta \min_s \theta(s)])\} \leq [W]_{i,1} \leq \max\{1, \exp[\beta \max_s \theta(s)])\}. \tag{66}$$

Using this, together with (53), (55), (56), (57), and (61), we showed that $\|z - c_1\mathbf{1}_S\|$ is upper bounded by a constant independent of $d$. This concludes the proof. $\qquad\square$

## A.9 BIAS ESTIMATES

**Lemma A.2.** *For any matrix $A$ and $\hat{A}$,*

$$\hat{A}^k - A^k = \sum_{h=1}^k \hat{A}^{h-1}(\hat{A} - A)A^{k-h}.$$

*Proof.* The proof follows from first principles:

$$\sum_{h=1}^k \hat{A}^{h-1}(\hat{A} - A)A^{k-h} = \sum_{h=1}^k \hat{A}^{h-1}\hat{A}A^{k-h} - \sum_{h=1}^k \hat{A}^{h-1}AA^{k-h} \tag{67}$$

$$= \sum_{h=1}^k \hat{A}^h A^{k-h} - \sum_{h=1}^k \hat{A}^{h-1}A^{k-h+1} \tag{68}$$

$$= \hat{A}^k - A^k + \sum_{h=1}^{k-1} \hat{A}^h A^{k-h} - \sum_{h=2}^k \hat{A}^{h-1}A^{k-h+1} \tag{69}$$

$$= \hat{A}^k - A^k. \tag{70}$$

$\square$

Henceforth, $\|\cdot\|$ will refer to $\|\cdot\|_\infty$, i.e. the induced infinity norm. Also, for brevity, we denote $\pi_{d,\theta}^C$ and $\hat{\pi}_{d,\theta}^C$ by $\pi_\theta$ and $\hat{\pi}_\theta$, respectively. Similarly, we use $d_{\pi_\theta}$ and $d_{\hat{\pi}_\theta}$ to denote $d_{\pi_{d,\theta}^C}$ and $d_{\hat{\pi}_{d,\theta}^C}$. As for the induced norm of the matrix $P$ and its perturbed counterpart $\hat{P}$, which are of size $S \times A \times S$, we slightly abuse notation and denote $\|P - \hat{P}\| = \max_s\{\|P_s - \hat{P}_s\|\}$, where $P_s$ is as defined in Section 2.

**Definition A.3.** Let $\epsilon$ be the maximal model mis-specification, i.e., $\max\{\|P - \hat{P}\|, \|r - \hat{r}\|\} = \epsilon$.

**Lemma A.4.** *Recall the definitions of $R_s, P_s, R_{\pi_b}$ and $P^{\pi_b}$ from Section 2, and respectively denote their perturbed counterparts by $\hat{R}_s, \hat{P}_s, \hat{R}_{\pi_b}$ and $\hat{P}^{\pi_b}$. Then, for $\epsilon$ defined in Definition A.3,*

$$\max\{\|R_s - \hat{R}_s\|, \|P_s - \hat{P}_s\|, \|R_{\pi_b} - \hat{R}_{\pi_b}\|, \|P^{\pi_b} - \hat{P}^{\pi_b}\|\} = O(\epsilon). \tag{71}$$

*Proof.* The proof follows easily from the fact that the differences above are convex combinations of $P - \hat{P}$ and $r - \hat{r}$. □

**Lemma A.5.** *Let $\pi_\theta$ be as in (5), and let $\hat{\pi}_\theta$ also be defined as in (5), but with $R_s, P_s, P^{\pi_b}$ replaced by their perturbed counterparts $\hat{R}_s, \hat{P}_s, \hat{P}^{\pi_b}$ throughout. Then,*

$$\|\pi_{d,\theta}^C - \hat{\pi}_{d,\theta}^C\| = O(\beta d \epsilon). \tag{72}$$

*Proof.* To prove the desired result, we work with (5) to bound the error between $R_s, P_s, P^{\pi_b}, R_{\pi_b}$ and their perturbed versions.

First, we apply Lemma A.2 together with Lemma A.4 to obtain that $\|(P^{\pi_b})^k - (\hat{P}^{\pi_b})^k\| = O(k\epsilon)$. Next, denote by $M$ the argument in the exponent in (5), i.e.

$$M := \beta[C_{s,d} + P_s(P^{\pi_b})^{d-1}\Theta].$$

Similarly, let $\hat{M}$ be the corresponding perturbed sum that relies on $\hat{P}$ and $\hat{r}$. Combining the bounds from Lemma A.4, and using the triangle inequality, we have that $\|\hat{M} - M\| = O(\beta d \epsilon)$.

Eq. (5) states that the C-SoftTreeMax policy in the true environment is $\pi_\theta = \exp(M)/(1^\top \exp(M))$. Similarly define $\hat{\pi}_\theta$ using $\hat{M}$ for the approximate model. Then,

$$\hat{\pi}_\theta = (\pi_\theta \circ \exp(M - \hat{M}))1^\top \exp(M)/(1^\top \exp(\hat{M})),$$

where $\circ$ denotes element-wise multiplication. Using the above relation, we have that $\|\hat{\pi}_\theta - \pi_\theta\| = \|\pi_\theta\|\|\frac{\exp(M-\hat{M})1^\top \exp(M)}{1^\top \exp(\hat{M})} - 1\|$. Using the relation $|e^x - 1| = O(x)$ as $x \to 0$, the desired result follows.

□

**Theorem A.6.** *Let $\epsilon$ be as in Definition A.3. Further let $\hat{\pi}_{d,\theta}^C$ being the corresponding approximate policy as given in Lemma 4.2. Then, the policy gradient bias is bounded by*

$$\left\|\frac{\partial}{\partial\theta}\left(\nu^\top V^{\pi_\theta}\right) - \frac{\partial}{\partial\theta}\left(\nu^\top V^{\hat{\pi}_\theta}\right)\right\| = \mathcal{O}\left(\frac{1}{(1-\gamma)^2}S\beta^2 d\epsilon\right). \tag{73}$$

We first provide a proof outline for conciseness, and only after it the complete proof.

*Proof outline.* First, we prove that $\max\{\|R_s - \hat{R}_s\|, \|P_s - \hat{P}_s\|, \|R_{\pi_b} - \hat{R}_{\pi_b}\|, \|P^{\pi_b} - \hat{P}^{\pi_b}\|\} = \mathcal{O}(\epsilon)$. This follows from the fact that the differences above are suitable convex combinations of either the rows of $P - \hat{P}$ or $r - \hat{r}$. We use the above observation along with the definitions of $\pi_{d,\theta}^C$ and $\hat{\pi}_{d,\theta}^C$ given in (5) to show that $\|\pi_{d,\theta}^C - \hat{\pi}_{d,\theta}^C\| = O(\beta d \epsilon)$. The proof for the latter builds upon two key facts: (a) $\|(P^{\pi_b})^k - (\hat{P}^{\pi_b})^k\| \leq \sum_{h=1}^{k}\|\hat{P}^{\pi_b}\|^{h-1}\|\hat{P}^{\pi_b} - P^{\pi_b}\|\|p^{\pi_b}\|^{k-h} = O(k\epsilon)$ for any $k \geq 0$, and (b) $|e^x - 1| = O(x)$ as $x \to 0$. Next, we decompose the LHS of (7) to get

$$\sum_s \left(\prod_{i=1}^4 X_i(s) - \prod_{i=1}^4 \hat{X}_i(s)\right) = \sum_s \sum_{i=1}^4 \hat{X}_1(s)\cdots\hat{X}_{i-1}(s)\left(X_i(s) - \hat{X}_i(s)\right)\times X_{i+1}(s)\cdots X_4(s),$$

where $X_1(s) = d_{\pi_{d,\theta}^C}(s) \in \mathbb{R}$, $X_2(s) = (\nabla_\theta \log \pi_{d,\theta}^C(\cdot|s))^\top \in \mathbb{R}^{S\times A}$, $X_3(s) = D(\pi_{d,\theta}^C(\cdot|s)) \in \mathbb{R}^{A\times A}$, $X_4(s) = Q^{\pi_{d,\theta}^C}(s,\cdot) \in \mathbb{R}^{A\times A}$, and $\hat{X}_1(s),\ldots,\hat{X}_4(s)$ are similarly defined with $\pi_{d,\theta}^C$ replaced by $\hat{\pi}_{d,\theta}^C$. Then, we show that, for $i = 1,\ldots,4$, (i) $\|X_i(s) - \hat{X}_i(s)\| = O(\epsilon)$ and (ii) $\max\{\|X_i\|, \|\hat{X}_i\|\}$ is bounded by problem parameters. From this, the desired result follows. □

*Proof.* We have

$$\frac{\partial}{\partial \theta}\left(\nu^\top V^{\pi_\theta}\right) - \frac{\partial}{\partial \theta}\left(\nu^\top V^{\pi'_\theta}\right) \tag{74}$$

$$= \mathbb{E}_{s \sim d_{\pi_\theta}, a \sim \pi_\theta(\cdot|s)}\left[\nabla_\theta \log \pi_\theta(a|s) Q^{\pi_\theta}(s,a)\right] - \mathbb{E}_{s \sim d_{\hat{\pi}_\theta}, a \sim \hat{\pi}_\theta(\cdot|s)}\left[\nabla_\theta \log \hat{\pi}_\theta(a|s) Q^{\hat{\pi}_\theta}(s,a)\right] \tag{75}$$

$$= \sum_{s,a}\left(d_{\pi_\theta}(s)\pi_\theta(a|s)\nabla_\theta \log \pi_\theta(a|s) Q^{\pi_\theta}(s,a) - d_{\hat{\pi}_\theta}(s)\hat{\pi}_\theta(a|s)\nabla_\theta \log \hat{\pi}_\theta(a|s) Q^{\hat{\pi}_\theta}(s,a)\right) \tag{76}$$

$$= \sum_s \left(d_{\pi_\theta}(s)(\nabla_\theta \log \pi_\theta(\cdot|s))^\top D(\pi_\theta(\cdot|s)) Q^{\pi_\theta}(s,\cdot)\right. \tag{77}$$

$$\left. - d_{\hat{\pi}_\theta}(s)(\nabla_\theta \log \hat{\pi}_\theta(\cdot|s))^\top D(\hat{\pi}_\theta(\cdot|s)) Q^{\hat{\pi}_\theta}(s,\cdot)\right) \tag{78}$$

$$= \sum_s \left(\prod_{i=1}^4 X_i(s) - \prod_{i=1}^4 \hat{X}_i(s)\right) \tag{79}$$

$$= \sum_s \sum_{i=1}^4 \hat{X}_1(s)\cdots\hat{X}_{i-1}(s)\left(X_i(s) - \hat{X}_i(s)\right)X_{i+1}(s)\cdots X_4(s), \tag{80}$$

where $X_1(s) = d_{\pi_\theta}(s) \in \mathbb{R}$, $X_2(s) = (\nabla_\theta \log \pi_\theta(\cdot|s))^\top \in \mathbb{R}^{S \times A}$, $X_3(s) = D(\pi_\theta(\cdot|s)) \in \mathbb{R}^{A \times A}$, $X_4(s) = Q^{\pi_\theta}(s,\cdot) \in \mathbb{R}^{A \times A}$, and $\hat{X}_1(s),\ldots,\hat{X}_4(s)$ are similarly defined with $\pi_\theta$ replaced by $\hat{\pi}_\theta$.

Therefore,

$$\left\|\frac{\partial}{\partial \theta}\left(\nu^\top V^{\pi_\theta}\right) - \frac{\partial}{\partial \theta}\left(\nu^\top V^{\pi'_\theta}\right)\right\| \le \left(\max_s \Gamma(s)\right) S, \tag{81}$$

where

$$\Gamma(s) = \|\sum_{i=1}^4 \hat{X}_1(s)\cdots\hat{X}_{i-1}(s)\left(X_i(s) - \hat{X}_i(s)\right)X_{i+1}(s)\cdots X_4(s)\|. \tag{82}$$

Next, since $d_{\pi_\theta}, d_{\hat{\pi}_\theta}, \pi_\theta$, and $\hat{\pi}_\theta$ are all distributions, we have

$$\max\{|X_1(s)|, |\hat{X}_1(s)|, |X_3(s,a)|, |\hat{X}_3(s,a)|\} \le 1. \tag{83}$$

Separately, using Lemma 4.3, we have

$$\|X_2\| = \|\nabla_\theta \log \pi_\theta(a|s)\| \le \beta(\|I_A\| + \|\mathbf{1}_A \pi_\theta^\top\|)\|P_s\|\|(P^{\pi_b})^{d-1}\|. \tag{84}$$

Since all rows of the above matrices have non-negative entries that add up to 1, we get

$$\|Y\| \le 2\beta. \tag{85}$$

In the rest of the proof, we bound each of $\|X_1 - \hat{X}_1\|, \ldots, \|X_4 - \hat{X}_4\|$.

Finally,

$$\|X_4\| \le \frac{1}{1-\gamma}. \tag{86}$$

Similarly, the same bounds hold for $\hat{X}_1, \hat{X}_2, \hat{X}_3$ and $\hat{X}_4$.

From, we have

$$\|X_1 - \hat{X}_1\| \le (1-\gamma)\sum_{t=0}^\infty \gamma^t \|\nu^\top (P^{\pi_\theta})^t - \nu^\top (P^{\hat{\pi}_\theta})^t\| \tag{87}$$

$$\le (1-\gamma)\|\nu\|\sum_{t=0}^\infty \gamma^t t d\epsilon \tag{88}$$

$$\le (1-\gamma)d\epsilon \sum_{t=0}^\infty \gamma^t t \tag{89}$$

$$= \frac{\gamma d\epsilon}{1-\gamma}. \tag{90}$$

The last relation follows from the fact that $(1 - \gamma)^{-1} = \sum_{t=0}^{\infty} \gamma^t$, which in turn implies

$$\gamma \frac{\partial}{\partial \gamma} \left( \frac{1}{1 - \gamma} \right) = \sum_{t=0}^{\infty} t \gamma^t. \tag{91}$$

From Lemma A.5, it follows that

$$\|X_3 - \hat{X}_3\| = O(\beta d \epsilon). \tag{92}$$

Next, recall that from Lemma 4.3 that

$$X_2(s, \cdot) = \beta \left[ I_A - \mathbf{1}_A (\pi_\theta)^\top \right] P_s \left( P^{\pi_b} \right)^{d-1}.$$

Then,

$$\|X_2(s, \cdot) - \hat{X}_2(s, \cdot)\| \leq \|\beta \left[ I_A - \mathbf{1}_A (\pi_\theta)^\top \right] P_s\| \| \left( P^{\pi_b} \right)^{d-1} - \left( \hat{P}^{\pi_b} \right)^{d-1} \| \tag{93}$$

$$+ \|\beta \left[ I_A - \mathbf{1}_A (\pi_\theta)^\top \right]\| \|P_s - \hat{P}_s\| \| \left( \hat{P}^{\pi_b} \right)^{d-1} \| \tag{94}$$

$$+ \beta \|\mathbf{1}_A (\pi_\theta)^\top - \mathbf{1}_A (\hat{\pi}_\theta)^\top\| \|\hat{P}_s \left( \hat{P}^{\pi_b} \right)^{d-1} \|. \tag{95}$$

Following the same argument as in (85) and applying Lemma A.2, we have that (93) is $O(\beta d \epsilon)$. Similarly, from the argument of (85), Eq. (94) is $O(\beta \epsilon)$. Lastly, (95) is $O(\beta d \epsilon)$ due to Lemma A.5. Putting the above three terms together, we have that

$$\|X_2(s, \cdot) - \hat{X}_2(s, \cdot)\| = O(\beta d \epsilon). \tag{96}$$

Since the state-action value function satisfies the Bellman equation, we have

$$Q^{\pi_\theta} = r + \gamma P Q^{\pi_\theta} \tag{97}$$

and

$$Q^{\hat{\pi}_\theta} = \hat{r} + \gamma \hat{P} Q^{\hat{\pi}_\theta}. \tag{98}$$

Consequently,

$$\|Q^{\pi_\theta} - Q^{\hat{\pi}_\theta}\| \leq \|r - \hat{r}\| + \gamma \|P Q^{\pi_\theta} - P Q^{\hat{\pi}_\theta}\| + \gamma \|P Q^{\hat{\pi}_\theta} - \hat{P} Q^{\hat{\pi}_\theta}\| \tag{99}$$

$$\leq \epsilon + \gamma \|P\| \|Q^{\pi_\theta} - Q^{\hat{\pi}_\theta}\| + \gamma \|P - \hat{P}\| \|Q^{\hat{\pi}_\theta}\| \tag{100}$$

$$\leq \epsilon + \gamma \|Q^{\pi_\theta} - Q^{\hat{\pi}_\theta}\| + \frac{\gamma}{1 - \gamma} \epsilon, \tag{101}$$

which finally shows that

$$\|X_4 - \hat{X}_4\| = \|Q^{\pi_\theta} - Q^{\hat{\pi}_\theta}\| \leq \frac{\epsilon}{(1 - \gamma)^2}. \tag{102}$$

$\square$

# B EXPERIMENTS

## B.1 IMPLEMENTATION DETAILS

The environment engine is the highly efficient Atari-CuLE (Dalton et al., 2020), a CUDA-based version of Atari that runs on GPU. Similarly, we use Atari-CuLE for the GPU-based breadth-first TS as done in Dalal et al. (2021): In every tree expansion, the state $S_t$ is duplicated and concatenated with all possible actions. The resulting tensor is fed into the GPU forward model to generate the tensor of next states $(S_{t+1}^0, \ldots, S_{t+1}^{A-1})$. The next-state tensor is then duplicated and concatenated again with all possible actions, fed into the forward model, etc. This procedure is repeated until the final depth is reached, for which $W_\theta(s)$ is applied per state.

We train SoftTreeMax for depths $d = 1 \ldots 8$, with a single worker. We use five seeds for each experiment.

For the implementation, we extend Stable-Baselines3 (Raffin et al., 2019) with all parameters taken as default from the original PPO paper (Schulman et al., 2017). For depths $d \geq 3$, we limited the tree to a maximum width of $1024$ nodes and pruned non-promising trajectories in terms of estimated weights. Since the distributed PPO baseline advances significantly faster in terms of environment steps, for a fair comparison, we ran all experiments for one week on the same machine and use the wall-clock time as the x-axis. We use Intel(R) Xeon(R) CPU E5-2698 v4 @ 2.20GHz equipped with one NVIDIA Tesla V100 32GB.

## B.2 TIME-BASED TRAINING CURVES

We provide the training curves in Figure 4. For brevity, we exclude a few of the depths from the plots. As seen, there is a clear benefit for SoftTreeMax over distributed PPO with the standard softmax policy. In most games, PPO with the SoftTreeMax policy shows very high sample efficiency: it achieves higher episodic reward although it observes much less episodes, for the same running time.

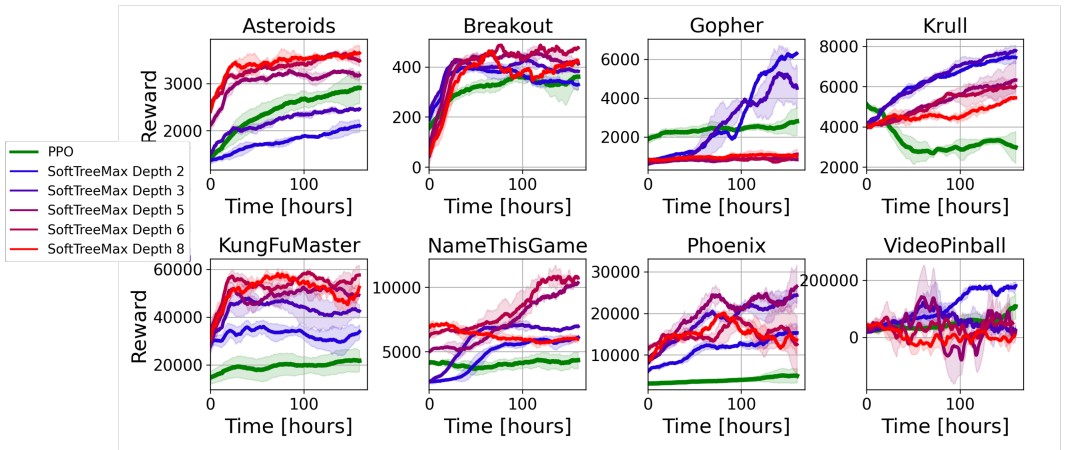

Figure 4: **Training curves: GPU SoftTreeMax (single worker) vs PPO (256 GPU workers).** The plots show average reward and standard deviation over 5 seeds. The x-axis is the wall-clock time. The runs ended after one week with varying number of time-steps. The training curves correspond to the evaluation runs in Figure 3.

## B.3 STEP-BASED TRAINING CURVES

In Figure 5 we also provide the same convergence plots where the x-axis is now the number of online interactions with the environment, thus excluding the tree expansion complexity. As seen, due to the complexity of the tree expansion, less steps are conducted during training (limited to one week) as the depth increases. In this plot, the monotone improvement of the reward with increasing tree depth is noticeable in most games.

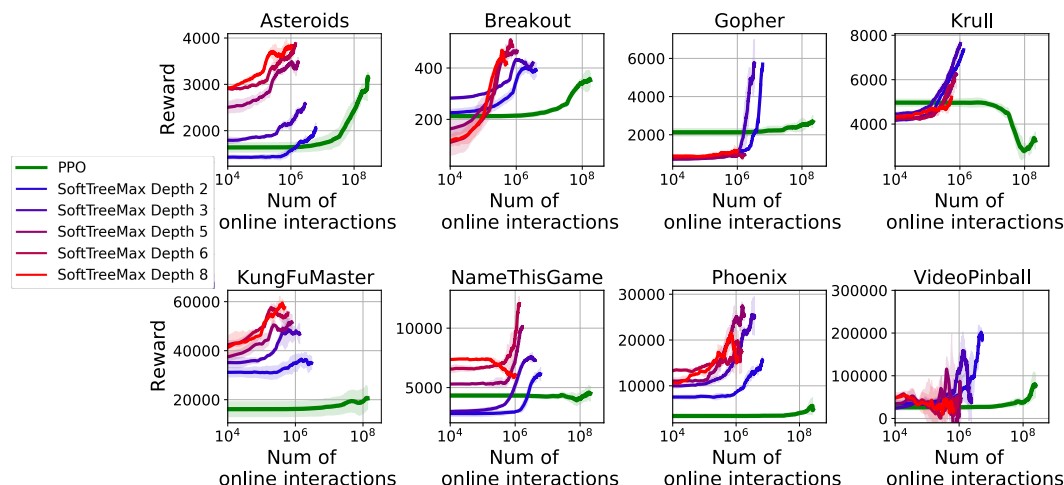

Figure 5: **Training curves: GPU SoftTreeMax (single worker) vs PPO (256 GPU workers).** The plots show average reward and standard deviation over 5 seeds. The x-axis is the number of online interactions with the environment. The runs ended after one week with varying number of time-steps. The training curves correspond to the evaluation runs in Figure 3.

We note that not for all games we see monotonicity. Our explanation for this phenomenon relates to how immediate reward contributes to performance compared to the value. Different games benefit differently from long-term as opposed to short-term planning. Games that require longer-term planning need a better value estimate. A good value estimate takes longer to obtain with larger depths, in which we apply the network to states that are very different from the ones observed so far in the buffer (recall that as in any deep RL algorithm, we train the model only on states in the buffer). If the model hasn't learned a good enough value function yet, and there is no guiding dense reward along the trajectory, the policy becomes noisier, and can take more steps to converge – even more than those we run in our week-long experiment.

For a concrete example, let us compare Breakout to Gopher. Inspecting Fig. 5, we observe that Breakout quickly (and monotonically) gains from large depths since it relies on the short term goal of simply keeping the paddle below the moving ball. In Gopher, however, for large depths (>=5), learning barely started even by the end of the training run. Presumably, this is because the task in Gopher involves multiple considerations and steps: the agent needs to move to the right spot and then hit the mallet the right amount of times, while balancing different locations. This task requires long-term planning and thus depends more strongly on the accuracy of the value function estimate. In that case, for depth 5 or more, we would require more train steps for the value to "kick in" and become beneficial beyond the gain from the reward in the tree.

The figures above convey two key observations that occur for at least some non-zero depth: (1) The final performance with the tree is better than PPO (Fig. 3); and (2) the intermediate step-based results with the tree are better than PPO (Fig. 5). This leads to our main takeaway from this work — there is no reason to believe that the vanilla policy gradient algorithm should be better than a multi-step variant. Indeed, we show that this is not the case.

## C  FURTHER DISCUSSION

### C.1  THE CASE OF $\lambda_2(P^{\pi_b}) = 0$

When $P^{\pi_b}$ is rank one, it is not only its variance that becomes 0, but also the norm of the gradient itself (similarly to the case of $d \to \infty$). Note that such a situation will happen rarely, in degenerate MDPs. This is a local minimum for SoftTreeMax and it would cause the PG iteration to get stuck, and to the optimum in the (desired but impractical) case where $\pi_b$ is the optimal policy. However,

a similar phenomenon was also discovered in the standard softmax with deterministic policies: $\theta(s, a) \to \infty$ for one $a$ per $s$. PG with softmax would suffer very slow convergence near these local equilibria, as observed in Mei et al. (2020a). To see this, note that the softmax gradient is $\nabla_\theta \log \pi_\theta(a|s) = e_a - \pi_\theta(\cdot|s)$, where $e_a \in [0, 1]^A$ is the vector with 0 everywhere except for the $a$-th coordinate. I.e., it will be zero for a deterministic policy. SoftTreeMax avoids these local optima by integrating the reward into the policy itself (but may get stuck in another, as discussed above).

