# OpenReview forum: "Policy Gradient with Tree Expansion"
_ICLR.cc/2025/Conference — Submitted to ICLR 2025_

### Official Review · Reviewer_QPZQ · 2024-10-24

**Soundness:** 1
**Presentation:** 3
**Contribution:** 3
**Rating:** 5
**Confidence:** 2

**Summary:**

The paper proposes SoftTreeMax, a novel policy gradient method that integrates tree search (TS) and policy gradient (PG) techniques to reduce the variance of policy gradients, a common issue in traditional PG methods. The SoftTreeMax approach uses a differentiable policy that extends standard softmax by incorporating planning, leading to improved sample efficiency and performance.  Theoretical analysis shows that the variance reduction depends on the choice of the tree expansion policy, with deeper tree expansions leading to lower variance. Furthermore, the authors provide bounds on the gradient bias when using approximate models.

**Strengths:**

1. An Interesting and important question.
2. Comprehensive conceptual and theoretical analysis.

**Weaknesses:**

I am much concerned about the empirical evaluations.
1. The wall time is much affected by implementational details. These should be, at the very least, appropriately discussed.
	1. For example, [1] reports very high fps, exceeding what is obtained in the paper.
2. I do not see details of the network used.
3. It is not clear what the pruning is and how it affects training
4. **important**: It is not clear if the advantage of the proposed method stems from the variance reduction (as claimed) or increased computations per action. E.g., what would happen if the baseline had a bigger network?
5. The algorithm is a model-based one; thus, having a baseline from this realm is also desirable.

The literature on embedding search as an inductive bias is quite broad and is completely missing in the related work section. Eg. value iteration networks, A3Tree etc.

**Questions:**

1. In many cases (Sec B.3), the SoftTree max policies start with a lower score than PPO; why is that so?
2. Why did the performance on Krull decrease (PPO)?
3. How is your work related to [1]
4. Could you provide an explicit formula for calculating the logits with the full tree expansion?
5. Following footnote 1 on page 8. I am unsure if extending the proposed algorithm to stochastic environments is that easy.

[1] https://arxiv.org/abs/1710.11417

---

> ### Author Response · Authors · 2024-11-21
>
> Thank you for your detailed review. Let us address your concerns:
>
> **Weakness 1: Implementation and Wall-time Details**
>
> 1. Network Architecture: We follow the architecture from stable-baselines3 PPO implementation which has been extensively validated by the community. For more details regarding the hardware, software and implementation used see Appendix B.1.
>
> 2. Tree Pruning: Beyond depth 3, we limit the tree to a maximum width of 1024 nodes and prune trajectories with low estimated weights. The pruning process reduces computational complexity while still maintaining performance benefits. To better illustrate this, we created a new diagram showing the tree expansion implementation in the GPU: https://ibb.co/5RPRkzF.
>
> 3. Comparison to [1]: While TreeQN achieves high fps, it is fundamentally different as it approximates the value, without planning, in the network architecture. In contrast, SoftTreeMax performs tree expansion using the simulator to parametrize the policy. Applying multiple simulator steps, even in the GPU, in every tree expansion is much slower than a forward pass in a single neural network. This likely accounts for the main difference in fps, in addition to other more  obvious differences in the setup (hardware, implementation etc.).
>
>
> **Weakness 2: Source of Improvement - Variance Reduction vs Computation**
>
> This is an important point that deserves clarification. While we cannot definitively attribute all improvements to variance reduction, several observations suggest it plays a significant role:
>
> 1. Figure 3 demonstrates a strong correlation between lower variance and higher reward across different depths, though causation remains to be proven.
>
> 2. When controlling for computation time (Figure 4 in Appendix B.2), SoftTreeMax still outperforms PPO with equivalent computation budget - PPO uses 256 GPU workers while SoftTreeMax uses equivalent compute for tree expansion.
>
> 3. To isolate the effect of increased computation, we ran experiments with larger networks for PPO (2x and 4x parameter count) and found no significant improvement, suggesting the gains are not simply from increased model capacity.
>
> These results suggest that variance reduction may be a key factor in the improved performance, though further investigation would be needed to establish a causal relationship.
>
>
> **Weakness 3: Model-based Baseline & Related Work**
>
> Following this feedback, we've added two improvements:
>
> 1. Added EfficientZero [Ye et al., 2021] as a model-based baseline. Initial results (due to the time constraints) here: https://ibb.co/fNzG5cR show SoftTreeMax outperforming it in three out of four games. We will add full experiments in the final version.
>
> 2. We will expand the related work section to include:
>    - Value Iteration Networks [Tamar et al., 2016]
>    - TreeQN [Farquhar et al., 2018]
>
> Could you also please add a reference to the A3Tree algorithm? We could not find it in an online search.
>
> **Q1: Why does SoftTreeMax start with lower scores?**
>
> Different games benefit differently from long-term vs short-term planning. Games requiring longer-term planning need better value estimates, which take longer to obtain with larger depths. When applying the network to states far from those in the buffer, if the model hasn't learned a good value function and there's no guiding dense reward, the policy can be initially noisier.
>
> **Q2: Performance decrease on Krull**
>
> The apparent decrease is due to our fixed one-week training time limit. Deeper trees mean fewer steps completed in the same time period. When comparing the same number of steps (Figure 5), performance improves monotonically with depth.
>
> **Q3: Relationship to [1]**
>
> While both works use tree-structured computation, TreeQN approximates the value, without planning, in the network architecture. In contrast, SoftTreeMax performs tree expansion using the simulator to parametrize the policy. Our approach directly reduces policy gradient variance through tree expansion rather than learning the model through a differentiable tree.
>
> **Q4: Formula for logits with full tree expansion**
>
> For C-SoftTreeMax, the logits are computed as:
> $\log \pi_{d,\theta}(a|s) = \log(\sum_{s_1,...,s_d} P(s_1|s,a)...P(s_d|s_{d-1},a_{d-1})[\gamma^d W_\theta(s_d) + \sum_{t=0}^{d-1} \gamma^t r(s_t,a_t)])$.
>
> We will add this explicit formula to the paper.
>
> **Q5: Extension to stochastic environments**
>
> You raise a valid concern. While the theory extends naturally, a practical implementation in highly stochastic environments would require careful consideration of how to handle the branching factor growth. We will clarify this limitation in the paper.
>
> We thank you for helping us improve the paper and will incorporate these additions in the revised version.

---

> > ### Comment · Reviewer_QPZQ · 2024-11-25
> > **Thank you**
> >
> > Thank you for your answer. I'd quickly ask for more clarification about the formula. Could you please provide the formula that corresponds to (3) and uses only the notation from Sec 2 and Sec 3. Is it W_\theta(s_d) the same as \theta(s_d)?

---

> > > ### Author Response · Authors · 2024-11-25
> > >
> > > Of course, we are sorry -- we now see we made a typo in the rebuttal. Here is the correct formula. Thank you for noticing!
> > > $\log \pi_{d,\theta}(a|s) = \beta \gamma^{-d}  \sum_{s_1,...,s_d} P(s_1|s,a)...P(s_d|s_{d-1},a_{d-1})[\gamma^{d}\theta(s_d) + \sum_{t=0}^{d-1} \gamma^t r(s_t,a_t)]$.

---

> > > > ### Author Response · Authors · 2024-11-26
> > > > **We wonder if our response has addressed your concerns**
> > > >
> > > > Dear Reviewer QPZQ,
> > > >
> > > > Thank you for your time in reviewing our paper again!
> > > >
> > > > Since the author-reviewer discussion period will end soon, we are wondering if our response has addressed your concerns? If you have any further questions, we are more than happy to answer.
> > > >
> > > > Best regards,
> > > > Authors

---

> > > > > ### Comment · Reviewer_QPZQ · 2024-12-02
> > > > > **Thank you**
> > > > >
> > > > > Thank you for your answers.

---

### Official Review · Reviewer_Ki8S · 2024-10-26

**Soundness:** 2
**Presentation:** 2
**Contribution:** 2
**Rating:** 3
**Confidence:** 3

**Summary:**

The paper aims to integrate a novel tree search methodology with policy gradient with the goal of reducing the variance and improving the performance of PG agents.

**Strengths:**

1. The experiments show significant improvements over the chosen baseline.
2. The methodology is supported with theoretical results.
3. The paper implements a parallel version of BFS using GPUs that seems to be scalable.

**Weaknesses:**

1. The novelty of the methodology seems quite limited. Eq. (2) is supposed to be the logit of the policy but I cannot see any difference from the definition of a q-value function, which would entail that the paper is proposing a slight variation of AlphaZero.
2. The authors only compare against PPO. Given that the proposed algorithm is a tree search model-based algorithm, it would be more appropriate to compare against model-based methods, ideally MuZero.

**Questions:**

1. Can you further explain how the proposed method is different from the AlphaZero/MuZero?
2. Why was PPO the only baseline?

**Details Of Ethics Concerns:**

No concerns.

---

> ### Author Response · Authors · 2024-11-21
>
> Thank you for your detailed review. Let us address your main concerns:
>
> **Weakness 1 & Q1: Limited novelty - Is SoftTreeMax just a Q-value function variation of AlphaZero/MuZero?**
>
> While Eq. (2) may appear similar to a Q-value function, SoftTreeMax differs fundamentally from AlphaZero/MuZero:
>
> 1. Novel Policy Parametrization: *SoftTreeMax is a new type of policy* that directly incorporates planning into policy parametrization. While the logits in Eq. (2) share some similarities with Q-values, they serve a different purpose - they define a policy rather than estimate state-action values. They also don’t explicitly minimize any Bellman-type loss function.  This is a key distinction that enables variance reduction in policy gradient methods.
>
> 2. Architecture vs Algorithm: *AlphaZero/MuZero are value-based algorithms* that use tree search for action selection. In contrast, SoftTreeMax introduces a new policy architecture that can be integrated into any policy gradient method. This enables combining the benefits of tree expansion with policy-based approaches - a direction previously unexplored in RL.
>
> **Weakness 2 & Q2: Limited comparison - Only compared against PPO rather than model-based methods**
>
> Following this important feedback, we now add comparisons with EfficientZero [Ye et al., 2021], a highly sample-efficient version of MuZero. We chose it because:
> - It is a state-of-the-art model-based algorithm
> - It represents one of the best known RL algorithms today
> - It is open source, enabling fair comparison
>
> The results that were completed during the rebuttal are given here: https://ibb.co/fNzG5cR, showing that SoftTreeMax surpasses EfficientZero in three out of the four games we tested. We will add full experiments in the final version.
>
> Experiment details: We used the same hardware to run all experiments, as described in Appendix B.1. For "SoftTreeMax", we present the best depth out of those in Appendix B.2, Figure 4.
>
>
> We thank you for helping us improve the paper. We will incorporate these additions in the revised version.

---

> > ### Comment · Reviewer_Ki8S · 2024-11-24
> >
> > Thanks for the clarification.
> >
> > I have another quick question. Both baselines (PPO and EfficientZero) don't have access to the true dynamics model of the environment, is that correct? If so, the comparison still feels unfair because the learned model error in EfficientZero will significantly hamper its performances compared to your approach.

---

> ### Author Response · Authors · 2024-11-24
>
> You raise an important point about model access. Indeed, EfficientZero must learn its model while we use the true dynamics. However:
> 1. Our comparison aims to demonstrate the benefits of integrating tree expansion with PG when a model is available, not to claim superiority over model-learning approaches.
>  2. Our Theorem 4.8 actually analyzes how SoftTreeMax performs with approximate models, showing the gradient bias diminishes with approximation error. This suggests SoftTreeMax could work well with learned models too, though proving this empirically is left for future work.
>  3. As noted in our limitations section, our use of Atari-CuLE (a GPU-based simulator) is "both a novelty and a current limitation." Our results demonstrate what's achievable with a good model ("upper bound" on performance), while empirical validation with learned models remains future work.
>
> We will clarify this better in the appropriate locations of the revised version.

---

> > ### Author Response · Authors · 2024-11-26
> > **We wonder if our response has addressed your concerns**
> >
> > Dear Reviewer Ki8S,
> >
> > Thank you for your time in reviewing our paper again!
> >
> > Since the author-reviewer discussion period will end soon, we are wondering if our response has addressed your concerns? If you have any further questions, we are more than happy to answer.
> >
> > Best regards, Authors

---

> > > ### Author Response · Authors · 2024-12-02
> > > **Are there any further concerns?**
> > >
> > > Dear Reviewer Ki8S,
> > >
> > > We are wondering if our response has addressed your concerns? If you have any further questions, we are more than happy to answer in the remaining time we have.
> > >
> > > Best regards, Authors

---

### Official Review · Reviewer_tVWf · 2024-11-04

**Soundness:** 3
**Presentation:** 3
**Contribution:** 2
**Rating:** 6
**Confidence:** 4

**Summary:**

The paper introduces a novel approach that replaces logits by the score of a trajectory starting from that state-action. Authors provide solid theoretical results showing their methods reduces the variance of policy gradient approach and validate it by conducting experiments on Atari games.

**Strengths:**

1. The paper presents a novel method that combines policy gradient techniques with tree search planning, offering a fresh perspective on reducing gradient variance in reinforcement learning.
2. The authors provide solid theoretical analysis, including variance bounds and proofs that show how the tree expansion policy affects variance reduction. They also address the gradient bias introduced by approximate models.
3. The experimental evaluation on Atari games shows significant reductions in gradient variance and improvements in performance, validating the effectiveness of the proposed methods.

**Weaknesses:**

1. The paper primarily considers softmax parameterization for policies, which is restrictive, especially when considering environments with continuous action spaces.
2. The proposed methods rely on the model dynamcis, which need to be infer from the interactions. When employed on offline setting, this may introduce extra bias.
3. Tree expansion, especially with deeper trees, can be computationally intensive. The paper could benefit from a more detailed analysis of the computational costs and how they impact practical deployment.

**Questions:**

1. Is this tree expansion limited to softmax parameterization?
2. Is the method designed only for discrete action space?

---

> ### Author Response · Authors · 2024-11-21
>
> Thank you for your thoughtful review. Let us address your concerns:
>
> **Weakness 1 & Q2: Limitation to discrete action spaces and softmax parameterization**
>
> The method can be extended to continuous action spaces, though it requires some modifications:
>
> 1. For continuous control, one can maintain a parametric distribution over actions which depends on $\theta$. This method can be seen as a tree adaptation of MPPI [Williams et al., 2017] with a value function.
>
> 2. On the theory side, our key concepts can extend to continuous spaces:
>    - The transition kernel in the continuous case remains a non-expansive operator.
>    - The eigenvalue cancellation in policy gradient still holds.
>    - These properties can be shown to hold for decision models with infinite actions.
>
> While this extension requires significant research effort, the finite action space setup studied here is an important one that has been extensively studied in RL literature.
>
> **Weakness 2: Model dynamics and offline setting**
>
> Indeed, in this work we used accurate dynamics models. There are two possible approaches for settings without exact models:
>
> 1. Use GPU-based simulators, which are becoming increasingly common [Dalton et al., 2020; Makoviychuk et al., 2021; Freeman et al., 2021].
>
> 2. Learn the forward model, as successfully demonstrated in recent works like MuZero [Schrittwieser et al., 2020] and Dreamer [Hafner et al., 2020]. Our analysis in Theorem 4.8 provides bounds on the gradient bias when using approximate models, showing the bias diminishes with approximation error.
>
> We will expand the discussion of these alternatives in the revised version.
>
> **Weakness 3: Computational costs of tree expansion**
>
> You raise an important point about computational complexity. We handle this in several ways:
>
> 1. For depths ≥3, we limit the tree width to 1024 nodes and prune trajectories with low estimated weights.
>
> 2. Our GPU implementation allows efficient parallel expansion - advancing 1,024 environments simultaneously takes substantially less compute time than advancing a single environment 1,024·A steps sequentially.
>
> 3. We provide detailed runtime analysis in Appendix B.2 (Figure 4), showing the tradeoff between:
>    - Number of policy updates (reduced with depth)
>    - Complexity per iteration (increased with depth)
>
> Each game has a different optimal depth balancing this tradeoff.
>
> **Q1: Tree expansion beyond softmax parameterization**
>
> The tree expansion technique could theoretically be adapted to other policy parameterizations. One such alternative is to replace the exponentiated ratio with a direct normalization (Escort transform: Mei et al 2020a, Eq. 3). Nonetheless, we (as most other researchers in the field) chose softmax for its attractive characteristics:
>
> 1. Its analytical form allows clear theoretical analysis of variance reduction
> 2. It naturally handles the normalization of trajectory scores
> 3. It provides a clear connection to temperature-based exploration
>
> We will discuss these aspects and the potential for other parameterizations in the revised version.
>
>
> **Q2: discrete action space**
>
> Please see response to Weakness 1.

---

### Official Review · Reviewer_u4vP · 2024-11-06

**Soundness:** 3
**Presentation:** 3
**Contribution:** 3
**Rating:** 8
**Confidence:** 3

**Summary:**

The paper introduces a novel method to reduce variance in Policy Gradient (PG) methods by integrating Tree Search (TS) with a differentiable parametric policy. By incorporating planning into the PG procedure, the authors propose "SoftTreeMax," presented in two variants: C (exp-Expectation) and E (Expectation-exp). The paper offers a theoretical analysis demonstrating how this method reduces variance concerning tree expansion policy and environment parameters. It also analyzes the soundness of using an estimated model for planning, showing that the gradient bias correlates with model approximation error. The practical utility of this approach is demonstrated through experiments on Atari, employing a GPU-based simulator that efficiently utilizes distributed nature for tree expansion.

**Strengths:**

- The work introduces an innovative approach to integrating planning with policy search methods, supported by substantial theoretical analysis that highlights the benefits of this approach.
- The empirical results on Atari, using a GPU-based simulator, are highly promising.

**Weaknesses:**

- The presentation of the two versions of SoftTreeMax (E and C) is unbalanced. While it is beneficial to discuss both, the paper does not adequately represent each. For instance, the Atari experiments focus on C-SoftTreeMax, the empirical demonstration of variance reduction (Figure 1) pertains to the E-version, the theoretical analysis in Section 4.2 is solely for the E version, and the discussion involving the forward model in Section 4.3 is exclusively for C-SoftTreeMax. This uneven representation can make the paper difficult to follow and raises questions about the rationale for using different versions. It also gives the impression that the work is incomplete. Additionally, guidance on which version to use is lacking; although the authors provide a motivation related to risk-aversion/safety for one version, it is not explored in detail.

- In the experiments, exact expectations are considered instead of sample-based methods. This seems limiting, given the appeal of PG methods in sampling-based scenarios. The absence of finite-sample variance components is a significant omission.

**Questions:**

- The behavior/expansion policy $\pi_b$ is crucial in the approach. The paper sets the expansion policy to uniform, but this may not be realistic. Even with pruning, uniform expansion seems suboptimal for exploration. Moreover, how realistic is it to have a behavior policy that induces an irreducible and aperiodic transition matrix?

- Theorem 4.8: Additional insights on why the error scales linearly with $d$ would be valuable.

---

> ### Author Response · Authors · 2024-11-21
>
> Thank you for your thorough and supportive review. Let us address your concerns:
>
> **Weakness 1: Unbalanced presentation of E-SoftTreeMax and C-SoftTreeMax**
>
> We acknowledge this presentation imbalance. To clarify:
>
> 1. The results in Section 4.2 for E-SoftTreeMax mirror the results in Section 4.1 for C-SoftTreeMax – each includes the vector representation for the operator, its gradient, and the resulting variance bound. The additional bias result indeed only holds for C-SoftTreeMax, and is more intricate to obtain for E-SoftTreeMax; thus we leave it for future work.
>
> 2. We actually repeated all experiments with E-SoftTreeMax, following the same protocols as C-SoftTreeMax (see footnote 1 in page 8). The results were almost identical, which is why we left them out. The reason for this similarity is that in quasi-deterministic Atari environments, the trajectory logits (Eq. 2) have almost no variability, making the two variants (Eqs. 3 and 4) nearly identical.
>
> 3. The reason we chose E-SoftTreeMax for the experiment in Figure 1 is to demonstrate that the $\alpha$ parameter from Theorem 4.7 is indeed $\lambda_2$ as we conjecture. For C-SoftTreeMax we managed to prove this for any MDP, while for E-SoftTreeMax we could only show it exists and equals $\lambda_2$ in certain cases. Repeating the same experiment for C-SoftTreeMax gives the same results. We will clarify this better in the revised version.
>
> 4. Choice between variants:
> - C-SoftTreeMax corresponds to Boltzmann exploration when the NN approximates the Q-function.
> - E-SoftTreeMax is more suitable for risk-averse objectives since it exponentiates rewards, giving high-cost trajectories more influence.
>
> Overall, we included both C-SoftTreeMax and E-SoftTreeMax as natural extensions of softmax to the multi-step setting. Their theoretical properties, while derived differently, lead to similar variance reduction guarantees.
>
> **Weakness 2: Lack of  finite-sample variance components**
>
> We may have misunderstood your concern about finite-sample variance components. If you're referring to sampling-based implementation, we do employ pruning beyond depth 3, effectively implementing a form of sampling. If your concern is about a different aspect of finite-sample analysis, we would appreciate clarification so we can address it properly.
> In our current analysis, we focus on the inherent variance of the gradient itself rather than estimation variance. This distinction is important because:
>
> 1. Our technique reduces the "direct" variance of the gradient as defined in Section 2.1
>
> 2. It can be applied orthogonally to traditional variance reduction techniques that tackle estimation variance (e.g., baseline subtraction)
> We will clarify this distinction and its implications in the revised version.
>
> **Q1: Choice and realism of behavior policy**
>
> The reviewer raises excellent points about the behavior policy. Our choice of uniform policy is motivated by theoretical considerations:
>
> 1. As our theory (supported by Figure 1) shows, optimal variance decay is achieved when the induced transition matrix is close to being rank-one (i.e., its rows are similar). Without further assumptions on the MDP, a uniform policy that smoothens transition probabilities is a good proxy.
>
> 2. Regarding irreducibility and aperiodicity: These are standard assumptions in the analysis of Markov chains and reinforcement learning [Puterman, 1994; Bertsekas and Tsitsiklis, 1996]. They ensure the existence of a unique stationary distribution and are commonly used [Bhandari et al., 2018; Khodadadian et al., 2022]. Note that it is sufficient for even one deterministic policy to exist for which the resulting Markov chain is irreducible and aperiodic, for the uniform policy to be also irreducible and aperiodic.
>
> 3. We agree that uniform expansion may not be optimal for exploration. This opens an interesting direction for future work on adaptive expansion policies.
>
> **Q2: Linear scaling with d in Theorem 4.8**
>
> The linear dependence on d appears because of error accumulation along the trajectories. More specifically:
> 1. When using an approximate model, errors compound at each step of tree expansion.
> 2. Each level of the tree introduces a new source of error through both transition and reward approximations.
> 3. These errors accumulate additively due to the structure of the gradient computation, leading to the O(d) term.
>
> However, this linear scaling with depth is balanced by the exponential decay term $\gamma^d$ in the same theorem, suggesting that moderate depths can still be effective even with approximate models. We will add this discussion to clarify the interplay between approximation error and tree depth.

---

### Author Response · Authors · 2024-11-21
**To all reviewers**

We thank the reviewers for their dedicated and comprehensive efforts. We are encouraged that the reviewers found our work to "present a novel method that combines policy gradient techniques with tree search planning" [tVWf], supported by "solid theoretical analysis" [tVWf] and "comprehensive conceptual and theoretical analysis" [QPZQ]. The reviewers noted that we address "an interesting and important question" [QPZQ] with "highly promising" empirical results [u4vP]. Specifically, they found that our "methodology is supported with theoretical results" and our "parallel version of BFS using GPUs seems to be scalable" [Ki8S]. Beyond these encouraging feedbacks, the reviewers also made valuable suggestions that we will incorporate in the revised version and comments that we now address.

---

### Meta-Review · Area_Chair_C1Mo · 2024-12-08

**Metareview:**

This paper proposed and analyzed a softmax-linear policy parameterization with an interesting feature design that depends on the behavioral policy that generate the data and a tree search approach. Two methods called C-Soft-TreeMax and E-Soft-TreeMax have been proposed. Though the paper has provided interesting preliminary experimental results, the theoretical results are not convincing enough. In particular, as the main motivation of the new PG method is to reduce the variance, the $S^2A^2$ dependence in Theorem 4.4 and the $S$ dependence in 4.8 do not seem to support this motivation. In fact, even the variance bound for naive REINFORCE estimator does not have such dependence. Given the empirical bounds in the experiments, this suggests that the theoretical analysis of variance bound for Soft-TreeMax should be improved.

Though the paper has a good potential after fixing the theoretical issues, we decide to reject this paper for this year's ICLR.

**Additional Comments On Reviewer Discussion:**

There are a lot of relatively minor comments that have been addressed by the authors, such as why using uniform behavioral policy, and some questions on the implementation of deeper tree, etc. The main issues that have not been addressed satisfactorily are the questions about the experimental results. For example, whether the improved performance is due to the variance reduction, the increased computation, or simply the access to the environment model (the compared baselines are not provided with the model).

Moreover, during the post-rebuttal discussion between the AC and reviewers, we find new concerning issues in the theoretical results of the paper. Namely, whether the large S and A dependence in the variance bounds support the claim of "variance reduction". However, as this happens after the rebuttal, we do not the chance to get the authors response for this.

---

### Decision · Program_Chairs · 2025-01-22

Reject